# Relationships between Extratropical Precipitation Systems and UTLS Temperatures and Tropopause Height from GPM and GPS-RO

**Benjamin R. Johnston** [1,2,*] , **Feiqin Xie** [2] **and Chuntao Liu** [2]

1   COSMIC Program Office, University Corporation for Atmospheric Research, Boulder, CO 80301, USA
2   Department of Physical and Environmental Sciences, Texas A&M University–Corpus Christi, Corpus Christi, TX 78412, USA; feiqin.xie@tamucc.edu (F.X.); chuntao.liu@tamucc.edu (C.L.)
*   Correspondence: bjohnston@ucar.edu

**Abstract:** This study characterizes the relationship between extratropical precipitation systems to changes in upper troposphere and lower stratosphere (UTLS) temperature and tropopause height within different environments. Precipitation features (PFs) observed by the Global Precipitation Measurement (GPM) satellite are collocated with GPS radio occultation (RO) temperature profiles from 2014 to 2017 and classified as non-deep stratospheric intrusion (non-DSI; related to convective instability) or deep stratospheric intrusion (DSI; related to strong dynamic effects on the tropopause). Non-DSI PFs introduce warming (up to 1 K) in the upper troposphere, transitioning to strong cooling (up to −3.5 K) around the lapse rate tropopause (LRT), and back to warming (up to 2.5 K, particularly over the ocean) in the lower stratosphere. UTLS temperature anomalies for DSI events are driven predominantly by large scale dynamics, with major cooling (up to −6 K) observed from the mid-troposphere to the LRT, which transitions to strong warming (up to 4 K) in the lower stratosphere. Small and deep non-DSI PFs typically result in a lower LRT (up to 0.4 km), whereas large but weaker PFs lead to a higher LRT with similar magnitudes. DSI events are associated with larger LRT height decreases, with anomalies of almost −2 km near the deepest PFs. These results suggest intricate relationships between precipitation systems and the UTLS temperature structure. Importantly, non-DSI PF temperature anomalies show patterns similar to tropical convection, which provides unification of previous tropical research with extratropical barotropic convective impacts to UTLS temperatures.

**Keywords:** GPS radio occultation; GPM; upper troposphere and lower stratosphere; lapse rate tropopause; extratropics; precipitation features; temperature anomalies

## 1. Introduction

The upper troposphere and lower stratosphere (UTLS) is a coupling region in the atmosphere, which is distinct in radiation, dynamics, chemistry, and microphysics [1,2]. A strong connectivity amongst these various processes makes the UTLS highly susceptible to climate change [3] and has attracted much research attention in recent decades. The UTLS is generally defined as the region ±5 km around the tropopause [4], and this boundary plays a crucial role in UTLS variability [5,6]. A fundamental characteristic of the tropopause is the change in static stability (temperature lapse rate) across the interface [7]. The WMO definition of the tropopause is based on lapse rate criteria (lapse rate tropopause or LRT) and is defined as "the lowest level at which the lapse rate decreases to 2 K km$^{-1}$ or less, provided that the average lapse rate between this level and all higher levels within 2 km does not exceed 2 K km$^{-1}$" [8]. The thermal tropopause provides a convenient way of defining the tropopause and is most widely used. There are also other ways of defining the tropopause, such as the potential vorticity-based dynamical tropopause [9] or the ozone-

based chemical tropopause [10]. Regardless of tropopause definition, it has been suggested that changes in tropopause height can serve as a useful indicator of climate change [3,11].

Stratosphere-troposphere exchange (STE) across the tropopause is an important bidirectional process influencing the chemistry of the UTLS [5]. Many recent studies have focused on understanding the role convection plays in STE throughout the tropics [12–14] since radiative-convective balance is the dominant physical process in the region [15]. Deep convection influences climate processes by regulating stratospheric water vapor through direct convective injection, by enhancing thin cirrus cloud presence, and by modulating the ozone budget in the upper troposphere [10,16,17]. However, understanding STE processes due to deep convection requires accurate temperature observations in the UTLS [18], so understanding the role convection plays in the heat budget of the tropical UTLS has been a prominent research topic over the past few decades. Warm anomalies have been observed throughout the mid-to-upper troposphere using radiosonde data [16] and climate models [1]. Cool anomalies have been observed in the tropical tropopause layer (TTL) using AIRS and radiosonde data [19] along with GPS radio occultation (RO) data [20–24]. The resulting temperature anomalies varied greatly among these studies, and the magnitude of these anomalies depended on surface type (land/ocean) and intensity of the deep convection [1,16,19–24].

On the other hand, the relationship of extratropical precipitation to UTLS temperature and tropopause height changes has attracted much less attention because of the significant challenge of quantifying the relative contributions of a variety of dynamical processes associated with extratropical STE [25]. These processes include tropopause folds near the subtropical and polar jets due to baroclinic wave dynamics [26,27], cutoff lows [28], gravity wave breaking [29], and mesoscale convective complexes [30]. For example, significant STE occurs due to turbulent mixing during baroclinic tropopause folds [31], and these folds can even reach nearly as low as the boundary layer [32]. Additionally, simulations have shown that a large fraction of STE near folds occurs due to precipitation below the depressions in the tropopause [33]. The diversity of weather systems that occur in the extratropics can also provide considerable challenges. Similar to the tropics, precipitation can form due to convective instability (such as from single cell thunderstorms and mesoscale convective complexes), but it also commonly occurs due to large-scale dynamics (such as within extratropical cyclones). Since these precipitation systems form in vastly different environments, characterizing their relationship to the UTLS thermodynamic structure is not as straightforward relative to the tropics. However, it is critical to study both synoptic and mesoscale extratropical precipitation systems and their relationship to UTLS temperatures and the tropopause to facilitate a better understanding of extratropical STE.

Space-borne radar observations from the Tropical Rainfall Measuring Mission (TRMM) have been useful in identifying the vertical extent of convection, but were restricted to the tropics and subtropics. Over the extratropics, most passive microwave and infrared satellite data lack the detailed vertical structure of storms and their associated vertical transport. Highly sensitive radar and lidar measurements from CloudSat and CALIPSO have aided the observations of vertical cloud structure in the extratropics. However, both satellites on the A-train orbit only capture storms at fixed local times of 1:30 P.M. and 1:30 A.M., which misses the peak occurrence of land convection in the extratropics [34]. Moreover, the small swath of both satellites limits the spatial sampling of individual storms. These gaps have been filled by the Global Precipitation Measurement (GPM) mission, which was launched in February 2014. The space-borne radar onboard the GPM satellite, along with high orbit inclination, extends the vertical scan of precipitation into the high latitudes [35]. Recent research utilizing GPM has enhanced our understanding of extratropical precipitation, including identifying the global distribution of storms with large sizes [36] and overshooting convection [37]. To better understand storm structure, additional information from nearby atmospheric thermodynamic profiles have historically been obtained from radiosonde soundings with limited spatial and temporal sampling or meteorological analyses/reanalyses with coarse vertical resolution and large uncertainty near the convec-

tion. GPS-RO soundings have filled these gaps by offering global observations of UTLS temperatures with a high vertical resolution in all-weather conditions [38].

The main goal of this research is to better understand the relationship between extratropical UTLS thermodynamic structure variations and the precipitation features (PFs) underneath. Specifically, extratropical PFs under two environments (e.g., Deep Stratospheric Intrusion (DSI) and non-Deep Stratospheric Intrusion (non-DSI)) are classified using potential vorticity and investigated separately because PF development is similar in nature within each environment, as non-DSI PFs form mainly due to convective instability, whereas PFs near DSI events typically form due to baroclinic instability.

To achieve this main goal, we have three major objectives: (1) To classify and establish the UTLS temperature and tropopause height anomaly pattern for non-DSI PFs. Since PFs within the non-DSI group are similar to tropical convection, we hypothesize that the UTLS anomalies should look relatively similar to previous results shown in the tropics [1,16,19–24]. Additionally, we want to quantify how PF depth, size, surface type, and season of occurrence can impact the sign and magnitude of UTLS temperature and tropopause height anomalies. Many studies [1,16,19–24] have attributed differences in the tropical convective temperature signal to those various factors, but this has yet to be analyzed in-depth for the extratropics. (2) To identify DSI events and determine the UTLS temperature and tropopause height anomalies that occur during these events. Most of the PFs that occur during these events are very different from tropical convection and are typically associated with extratropical cyclones or strong frontal boundaries. Thus, we hypothesize that the anomaly patterns should look different from tropical convection and will be strongly influenced by synoptic-scale extratropical dynamics. (3) To determine where and when the two populations of PFs occur and what their properties are, as this knowledge will aid in providing physical explanations for the observed anomalies.

In this study, extratropical PFs are identified using GPM, while the temperature structure within these PFs is provided by collocated high-resolution GPS-RO soundings. The structure of this paper is as follows: Section 2 introduces the GPM, GPS, and ERA-Interim data used in this study; Section 3 describes the methodology used, including how the background profiles are generated and how the PFs are separated into two groups; Section 4 provides sampling for PFs in both environments along with PF characteristics; Section 5 presents the key results of the study, including UTLS temperature and tropopause height anomalies within each environment; Lastly, a discussion, including the main conclusions and study limitations, is provided in Section 6.

## 2. Data Description

### 2.1. GPM Precipitation Feature Product

The GPM mission is an international network of satellites that provide global observations of precipitation. Building upon the many successes of TRMM, which focused primarily on quantifying the three-dimensional distribution of moderate-to-heavy rain throughout the tropics [39], the advanced radar/radiometer system onboard the GPM Core Observatory extends the measurement range to include light precipitation and snow. The GPM Core Observatory is equipped with the first spaceborne dual-frequency phased array precipitation radar (DPR), which operates at the Ku and Ka bands (13 and 35 GHz, respectively), and a conical-scanning multichannel (10–183 GHz) microwave imager (GMI) [35]. The 65° orbit inclination allows for observations into the high latitudes where much of the precipitation has a lighter intensity. GPM horizontal resolution is 5 km × 5 km and Ku-band vertical resolution is 250 m. In this study, GPM radar PFs are obtained from 2014 to 2017. The GPM PF database uses an algorithm similar to the TRMM PF database originally developed at the University of Utah [40]. The PFs are defined by grouping contiguous areas with nonzero near-surface precipitation using the Ku-band radar [41], and the location (lat/lon) of the PF is the centroid of a best-fit ellipse. This analysis method condenses the original pixel-level measurements into the properties of events, which greatly increases the efficiency of searching and sorting the observed historical events [40]. Maximum echo-top

heights for calculating PF depth are obtained using the GPM KuPR, which has a minimum detectable reflectivity near 12 dBZ [42]. For some PFs with large areas, the PF centroid and maximum echo-top height location could be quite different (most of these PFs are elongated fronts with continuous precipitation). Therefore, PFs with >2° lat/lon difference between the centroid and maximum echo-top height are excluded from this study, which removes 1132 PFs (0.16%) prior to collocation.

## 2.2. GPS Radio Occultation Data

GPS-RO soundings are obtained from three missions for this study: the joint US-Taiwan six-satellite FORMOSA Satellite Series No. 3/Constellation Observing System for Meteorology, Ionosphere, and Climate (COSMIC) mission [38], the German TerraSAR-X satellite [43], and the GRACE-B satellite [44]. The COSMIC constellation provided over 1000 soundings per day with relatively homogeneous sampling coverage around the globe when GPM was launched in early 2014. However, the number of daily soundings decreased to roughly 250 per day at the end of 2017. To increase the sampling numbers, additional GPS-RO observations from the TerraSAR-X and GRACE-B satellites are obtained. Both satellites provided ~250 soundings/day and comprise ~25% of the total collocations with GPM. The reprocessed level-2 RO profiles for all three missions are obtained from the COSMIC Data Analysis and Archive Center (CDAAC) at the University Corporation for Atmospheric Research (UCAR). The profiles are quality controlled by excluding the ones with "bad" flags (such as if the observation bending angles exceed the climatology by a specific threshold). We use the "atmPrf" product, which provides refractivity and dry temperature ($T_{dry}$) from usually near the surface up to ~60 km. At microwave frequencies, the atmospheric refractivity $N$ is related to atmospheric pressure $P$, temperature $T$, and water vapor partial pressure $e$ [45]:

$$N = k_1 \frac{P}{T} + k_2 \frac{e}{T^2} \tag{1}$$

where $k_1$ is 77.6 K hPa$^{-1}$ and $k_2$ is 3.73 × 10$^5$ K$^2$ hPa$^{-1}$. The dry temperature is derived from the refractivity Equation (1) by neglecting atmospheric moisture [46], the second term within (1), such that:

$$T_{dry} = k_1 \frac{P_{dry}}{N} \tag{2}$$

where $P_{dry}$ is the dry pressure derived through hydrostatic integration by again neglecting the moisture within (1). In this study, dry temperature is used because it can be treated as an independent satellite retrieval, whereas the real temperature retrieval ("wetPrf") relies on a priori moisture information from the European Centre for Medium-Range Weather Forecasts (ECMWF) low-resolution analysis. Note, however, the dry temperature retrieval is nearly identical to the real temperature in the UTLS region, as moisture is generally negligible when temperatures are lower than 250 K [47]. Thus, only sounding altitudes with a $T_{dry}$ less than 250 K were used in this study to minimize the impact of water vapor on the temperature [24]. Therefore, the RO dry temperature in the UTLS region is denoted as RO temperature as it represents the real atmospheric temperature. The retrieved profiles are reported as a function of geometric height above mean sea level and the location of each profile nearest to the surface is used. The vertical resolution of RO soundings varies from 0.2 km in the lower troposphere to 1.4 km in the upper stratosphere [38] with an average of 0.5 km in the UTLS [48], while the horizontal footprint of tangent point trajectories is ~200 km [38]. Consistency and structural uncertainty of RO temperatures are similar between the various missions and uncertainties have been demonstrated to be smallest in the UTLS region between 8–25 km [49].

The monthly zonal mean (using 2.5° grids) global temperatures, temperature standard deviation, and LRT height are shown in Figure 1 using COSMIC profiles from 2006–2017. Throughout the tropics, LRT heights are consistently high year-round (~15 to 17 km) and UTLS temperature standard deviation remains low. A transition region occurs in the

midlatitudes around 30° along with a sharp "tropopause break" near the subtropical jet, when the tropical tropopause extends to higher latitudes and overlies the extratropical lower stratospheric air. This transition region migrates with season. In the wintertime, it moves equatorward, becomes sharper and, consequently, the temperature standard deviation increases, whereas in the summer, it moves poleward and the tropopause height decreases more gradually. Several temperature standard deviation "hotspots" are evident throughout the UTLS, such as during the Northern Hemisphere (NH) summer monsoon (JJA) around ~30° N when deep convection frequently occurs. In the high latitudes, the tropopause remains low year-round (~8 to 12 km) with a large temperature standard deviation that reaches a maximum during the winter. The extremely large standard deviation throughout the polar lower stratosphere is due to variability along the edges of the polar vortex. The Arctic vortex reaches its highest variability during the NH winter (DJF), while the Antarctic vortex displays its largest variability in the Southern Hemisphere (SH) spring (SON) since it is stronger, larger, and longer-lasting [50]. The large standard deviation of UTLS temperatures observed throughout the mid-to-high latitudes is of particular interest in this study. In the tropics, diabatic heating from frequent deep convection is the dominant process, yet temperature standard deviation remains relatively small year-round. However, in the extratropics, both moist and dry dynamics play a role in the development of precipitation and contribute to the large temperature standard deviation. Thus, proper characterization of extratropical environments is paramount in accurately determining how PFs relate to their environment and impact UTLS temperature variation.

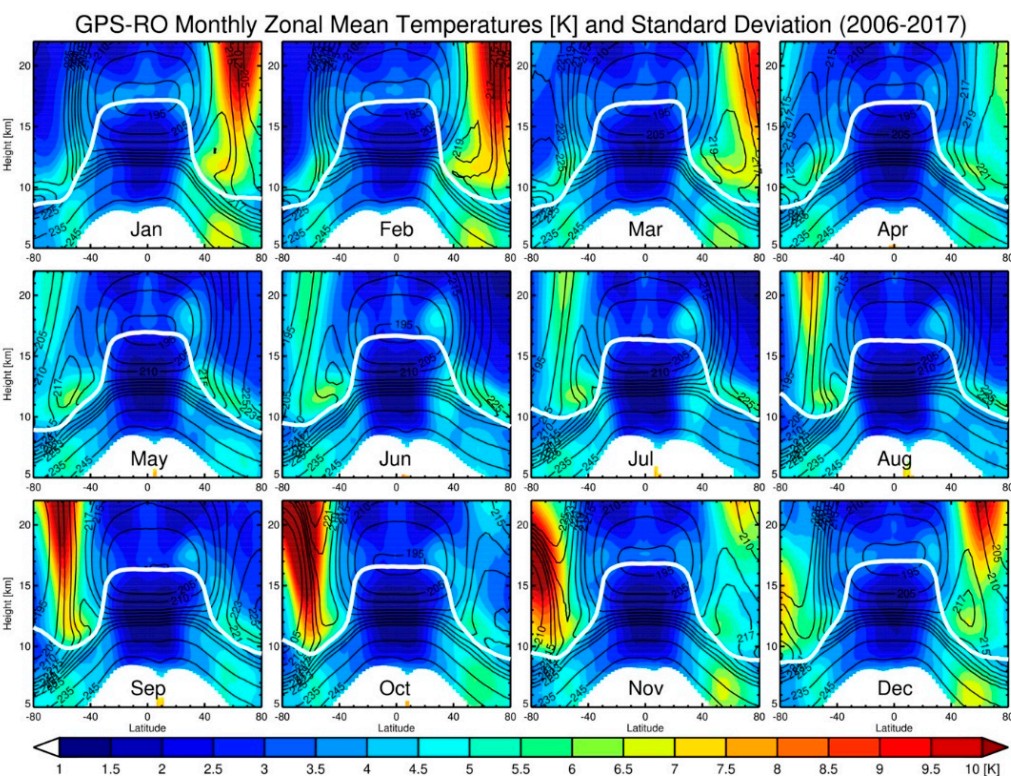

**Figure 1.** Climatology of UTLS monthly zonal mean temperatures (solid contours, in K) and temperature standard deviation (color contours, in K) between 80° N and 80° S derived from COSMIC GPS-RO profiles from 2006 to 2017. Lapse-rate tropopause height is shown with a solid white contour. Temperatures greater than 250 K are removed. Additional temperature contours are added between 205–225 K to better display UTLS structure.

### 2.3. ERA-Interim Data

ERA-Interim is a global reanalysis produced by the ECMWF. This gridded dataset covers 1979–2019 in six-hour intervals [51]. Pressure-level potential vorticity data are used,

which have a horizontal resolution of 0.75° latitude × 0.75° longitude with 37 vertical levels from 1000 to 1 hPa. Roughly six to seven levels are provided within the UTLS.

## 3. Methodology

To analyze the relationship between PFs and the thermodynamic structure of the UTLS, GPS-RO temperature profiles are collocated with PFs observed by GPM from 2014 to 2017 throughout the extratropics (20–65°). The collocated GPS profile must be within 3 h after PF occurrence and within a 300 km radius of the PF. Note that the 3 h and 300 km collocation criteria are chosen to provide a good balance between robust sampling and observation of the direct impact to the PF's surrounding environment. In addition, for each RO temperature profile, the nearest-neighbor ERA-I potential vorticity profile is obtained.

Throughout the extratropics, considerable seasonal and meridional variation of the local tropopause height is observed [52], and the top of the troposphere coincides closely to the level above which convection rarely penetrates. This is because the upper bound of tropospheric clouds is strongly constrained by radiative cooling from water vapor and the depth of tropospheric mixing, and these physical linkages have been applied to both tropical and extratropical convection [53]. Therefore, to account for the large variation of tropopause heights, we introduce the PF relative depth (RD), which measures the fractional PF depth (e.g., the maximum echo-top height) relative to the local LRT height:

$$\text{RD} = \frac{Maximum\ Height\ of\ Detectable\ Radar\ Echoes}{Lapse\ Rate\ Tropopause\ Height} \tag{3}$$

We use a relative definition of depth because the absolute difference does not take into account the "size" of the quantities involved (e.g., the absolute difference of an echo-top height of 4 km and LRT height of 9 km is the same as an echo-top height of 12 km and LRT height of 17 km). Since we focus on PFs that reach near/into the UTLS, only PFs with a maximum echo-top height of at least 50% of the LRT height (e.g., RD ≥ 0.5) are studied.

### 3.1. PF and GPS Temperature Profile Classification Using Potential Vorticity

Over the tropics, PFs typically result from convective instability. However, over the extratropics, PFs can develop not only from tropical-like convective instability, but also baroclinic instability. Since the environments these PFs develop in can be drastically different, it is necessary to devise a method of classifying the PFs that form in these two different environments. In the extratropics, potential vorticity (PV) is a tracer-like variable based on the conservation of thermodynamic properties and momentum. It has long been used to identify the dynamical tropopause, as the stratosphere is characterized by significantly higher values of PV [9,54,55]. The extratropical tropopause is remarkably close to the ±2-PVU surface, where PVU denotes the standard potential vorticity unit (1 PVU = $10^{-6}$ m$^2$ s$^{-1}$ K kg$^{-1}$) [5]. PV on various isentropic surfaces has often been used to identify the intrusion of stratospheric air into the troposphere. Namely, the intersection of the 320 K isentrope to the 2-PVU surface has been used to identify moderate to deep stratospheric intrusions or tropopause folds, which frequently occur near extratropical cyclones during a strong frontal passage from late fall through early spring [25,56–58]. It has also been suggested that the intrusion of stratospheric air may impact precipitation development, as the convective environment is often enhanced when drier stratospheric air descends to the middle levels [59,60].

In this study, we use PV on the 320 K isentrope to classify the observed PFs into two categories where the trigger for PF development is similar in nature within the environment (baroclinic instability vs. convective instability). All GPS profiles with PV of at least 2-PVU on the 320 K isentrope (identified from collocated ERA-I PV) are labeled DSI as the associated PFs coincide with strong dynamic effects on the tropopause through the aforementioned stratospheric intrusions. Most of the PFs within this classification occur due to large baroclinic instability and are found within extratropical cyclones, cut-off lows, or along strong frontal boundaries. These PFs occur most frequently in fall, winter, and

spring throughout the mid/high latitudes. On the other hand, profiles with PV of less than 2-PVU at 320 K are labeled non-DSI as the associated PFs are more likely to be related to convective instability. The PFs within this group mainly include a variety of weather phenomena that occur most frequently from mid-spring through early fall, such as single cell thunderstorms, supercells, squall lines, mesoscale convective systems, and tropical cyclones. After classifying the GPS profiles, non-DSI and DSI median monthly background temperatures/tropopause heights are derived using profiles within 2.5° latitude × 5° longitude grids. Note that RO profiles from 2006–2017 are used to derive the background profiles and the grid sizes were chosen to allow for enough RO profiles in each grid cell. Additionally, a minimum of 30 profiles are required in a grid cell for deriving the median background profile for either environment, and the median statistical method was chosen to reduce the impact of outliers on the background profiles. Since we use monthly gridding, the presence of both environments can exist within one grid during that month (e.g., as different synoptic patterns propagate over the midlatitudes during winter). To reduce the uncertainty of the PF classification solely based on PV values, the simple least-squares method is used to determine the PF's classification by computing the median absolute deviation (MAD) of the PF temperature profile and the two median background profiles within ±5 km around the PF LRT height. The MAD was used in this scenario since it is more resistant to outliers than the standard deviation. Then, the background profile with a smaller MAD is the classification assigned to the PF, which makes the PF classification more robust.

Figure 2 displays the seasonal GPS-RO sampling for both non-DSI (blue contours) and DSI (red contours) background profiles along with gridded median LRT height differences (LRT$_{\text{non-DSI}}$–LRT$_{\text{DSI}}$, color shaded) for the transition regions where both types of PFs coexist. The number of GPS profiles in each grid generally ranges from 60 to 120 in areas with one type of PF, but can be significantly reduced in areas where both environments occur (e.g., the midlatitudes). The transition region shifts equatorward as winter approaches and moves poleward in the summer. The corresponding LRT height differences between the median non-DSI and DSI backgrounds also shows large seasonal variation, especially in the NH due to larger land/ocean contrast. The largest differences are typically observed over the oceans. In January (Figure 2a), height differences of 2.5 to 3.5 km are seen over the Kuroshio and Gulf Stream warm ocean currents. This can be attributed to consistent changes in air mass type and large baroclinicity along the coasts. On the other hand, the smallest variability occurs over the continental interior as conditions are frequently cold and dry. The transition region shifts much farther north in July (Figure 2c), with few DSI profiles and small LRT height differences. In general, the transition region coincides well with the latitude band showing the largest UTLS temperature variation (between 40–60° from Figure 1), which further confirms the importance of separating the two distinct PF categories using PV for an extratropical UTLS study.

### 3.2. Additional Quality Control

In addition to the large tropopause height standard deviation within 40–60°, there can be large intraseasonal LRT height variations throughout the subtropics (~25–35°), especially during the winter to early spring months in both hemispheres where standard deviations often exceed 2 km. For example, studies have shown a bimodal distribution of subtropical tropopause heights, with one mode above 15 km (characteristic of the tropical tropopause) and another one below 13 km (typical of the extratropical tropopause) that has been shown to be related to the formation of multiple tropopauses [61,62]. Pan et al. [63] has also shown the occurrence of multiple tropopauses extending to nearly 60° N. Therefore, in order to minimize the impact that intraseasonal LRT height variation could have on monthly median GPS background temperature profiles and anomaly calculations, PFs that occur within any grids with monthly LRT height standard deviations exceeding 2 km are excluded from this study (which removes ~6.9% of PFs).

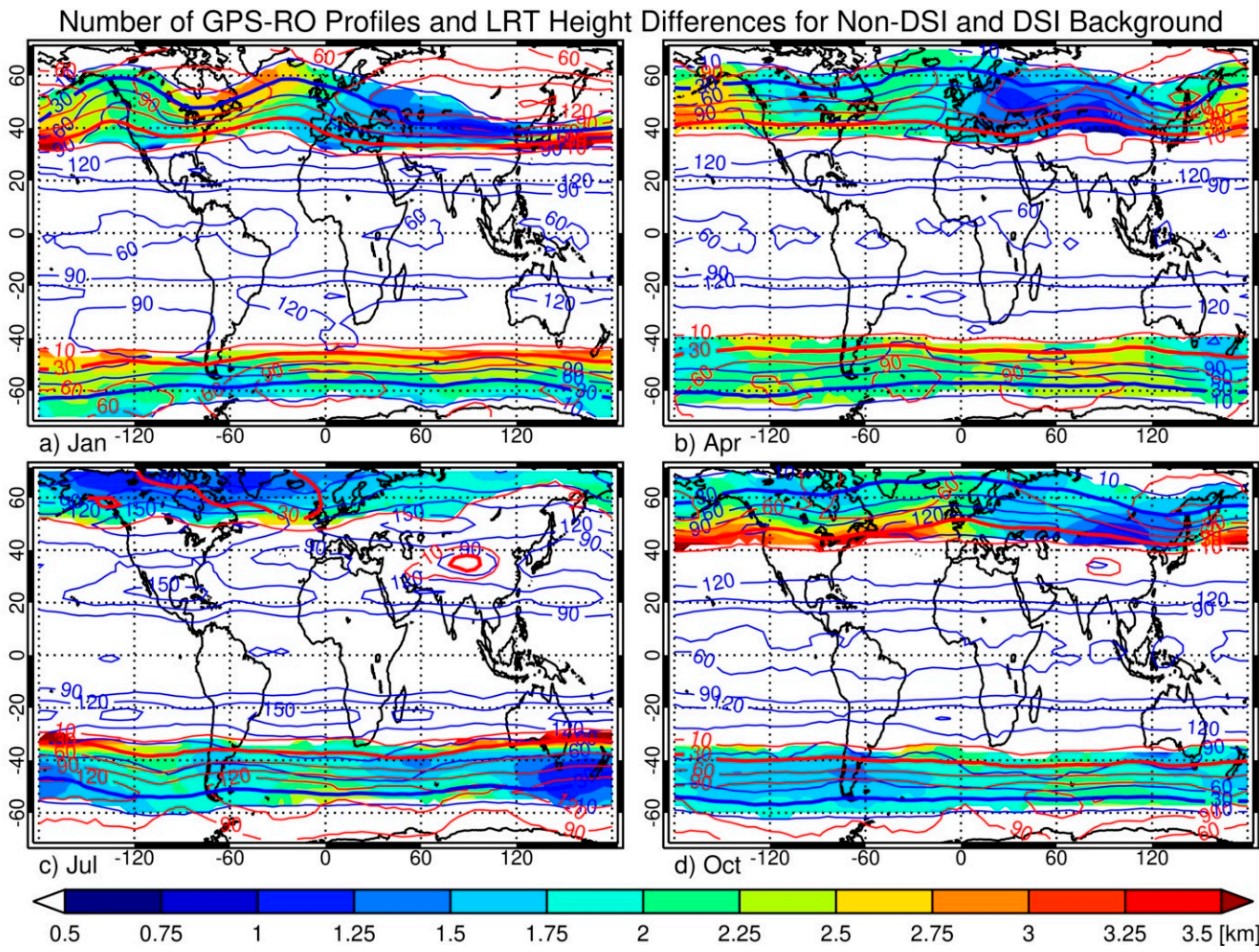

**Figure 2.** Number of GPS-RO profiles for non-DSI (solid blue contours) and DSI (solid red contours) backgrounds from 2006 to 2017 for (**a**) January, (**b**) April, (**c**) July, and (**d**) October. Shaded contours display LRT height differences (km) between non-DSI and DSI background profiles, provided the sampling is at least 10 profiles within each grid.

Note that there also may be a PF-type ambiguity within larger DSI PFs. For example, many large-size DSI PFs form along elongated cold fronts in the wintertime with a large meridional and small zonal extent. If the pixels identified by the GPM KuPR are contiguous along the length of the front, then the front would be considered a single PF. However, there may be some deeper convective features near the equatorward-end of the PF, which may be some distance away from the stratospheric intrusion. While steps are taken to minimize this limitation (such as filtering out PFs with a distance greater than 2° between the PF center and the location of PF maximum height, see Section 2.1), some stronger convective features with more non-DSI characteristics may remain within large DSI PFs.

### 3.3. UTLS Temperature Anomaly and Tropopause Height/Temperature Anomaly Calculations

In this study, each individual PF temperature anomaly profile ($T'$) is derived by subtracting the respective non-DSI or DSI gridded median GPS background temperature ($T_{BG}$) profile from the GPS PF temperature profile ($T_{PF}$):

$$T'(z) = T_{PF}(z) - T_{BG}(z) \tag{4}$$

where height $z$ is from within $\pm 5$ km of the PF's LRT height to account for the large seasonal and meridional variations in tropopause height. Figure 3 shows two examples of GPS temperature profiles collocated with PFs (red) and their gridded median non-DSI and DSI background profiles (black). Figure 3a shows a relatively large-size (20,229 km$^2$)

overshooting non-DSI PF (RD = 1.07). Overshooting (or tropopause-penetrating) PFs are generally considered some of the most intense convective systems on Earth and have attracted much attention with regards to STE [64]. The temperature anomaly profile (Figure 3c) is calculated by subtracting the non-DSI background profile (solid black line) from the PF profile. The temperature anomalies for each PF are centered on the PF's LRT height and obtained for a ±5 km window around this height. Then, the anomalies are grouped according to the different PF characteristics (PF size and RD) and averaged to generate a mean anomaly profile for each group. The classification of the PF temperature profile from Figure 3a is relatively straightforward. However, there are some instances (e.g., Figure 3b) that can be somewhat ambiguous. While the upper tropospheric section of the PF profile is certainly "non-DSI" in nature, the tropopause/lower stratosphere is rather uncertain. Thus, the temperature anomaly profile from Figure 3d looks quite different than Figure 3c due to the PF classification uncertainty.

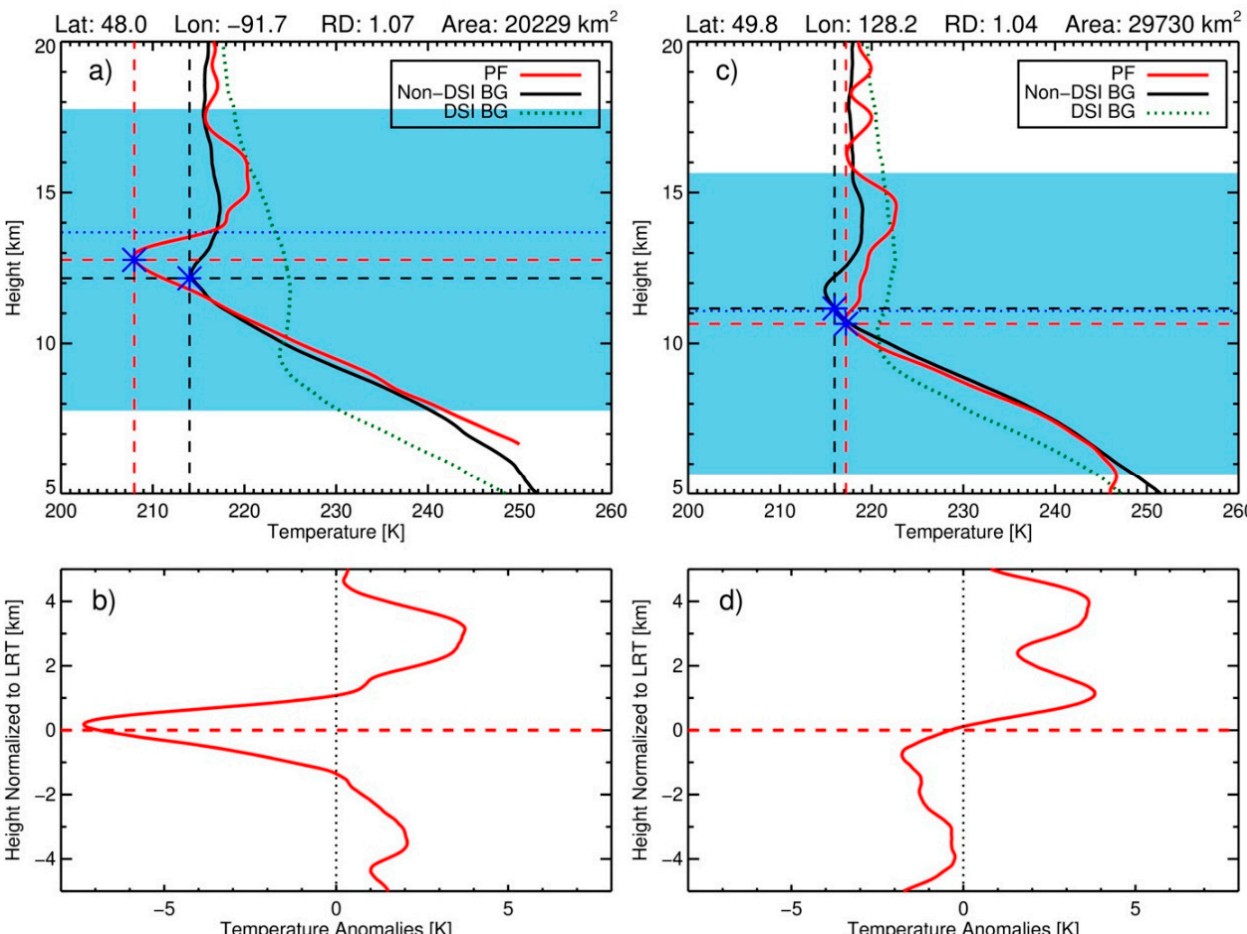

**Figure 3.** (**a**,**b**) Two overshooting non-DSI PFs with collocated GPS temperature profiles (solid red) along with their associated non-DSI (solid black) and DSI (dotted green) background profiles. Blue asterisks and the associated dashed lines indicate the height and temperature of the background and PF LRT. Dotted blue line indicates the PF maximum echo-top height. Blue-shaded area shows the region ±5 km around the PF's LRT height for anomaly calculation. Temperature anomaly profiles are shown on the bottom (**c**,**d**) with the "zero" height corresponding to the PF's LRT height.

Similarly, LRT height/temperature anomalies are also computed by subtracting the median background from the PF tropopause height/temperature:

$$\mathrm{LRT}H' = \mathrm{LRT}H_{\mathrm{PF}} - \mathrm{LRT}H_{BG} \tag{5}$$

$$\mathrm{LRT}T' = \mathrm{LRT}T_{\mathrm{PF}} - \mathrm{LRT}T_{BG} \tag{6}$$

The PFs are again categorized by their area and RD to determine the relationship between PF characteristics and anomaly magnitude.

It is worth noting the challenge in classifying PFs in the extratropics for a composite study such as this. For example, when using PV, any potential temperature surface chosen (e.g., 320 K) will intersect the tropopause at different latitudes throughout the year [25]. However, the joint distribution of LRT height/temperature anomalies derived from collocated GPS profiles and PV values for extratropical PFs clearly shows two distinct PF categories separated by the ±2-PVU threshold at 320 K (Figure 4). Note that positive PVU values indicate PFs within the NH and negative PVU values indicate PFs within the SH. Non-DSI PFs have most samples clustered to minor LRT anomalies, although a few LRT height/temperature anomalies can reach ±5 km/±20 K. However, there is a distinct change in the anomalies observed around ±2-PVU. The majority of DSI events display negative LRT height anomalies along with a corresponding LRT temperature increase. The anomalies are generally larger in the SH than the NH. Additionally, the anomaly magnitudes tend to become larger as PVU values also become larger. These clear differences in tropopause anomalies support how the PV method can successfully separate the PFs into two general groups that have distinct opposite influences on the tropopause.

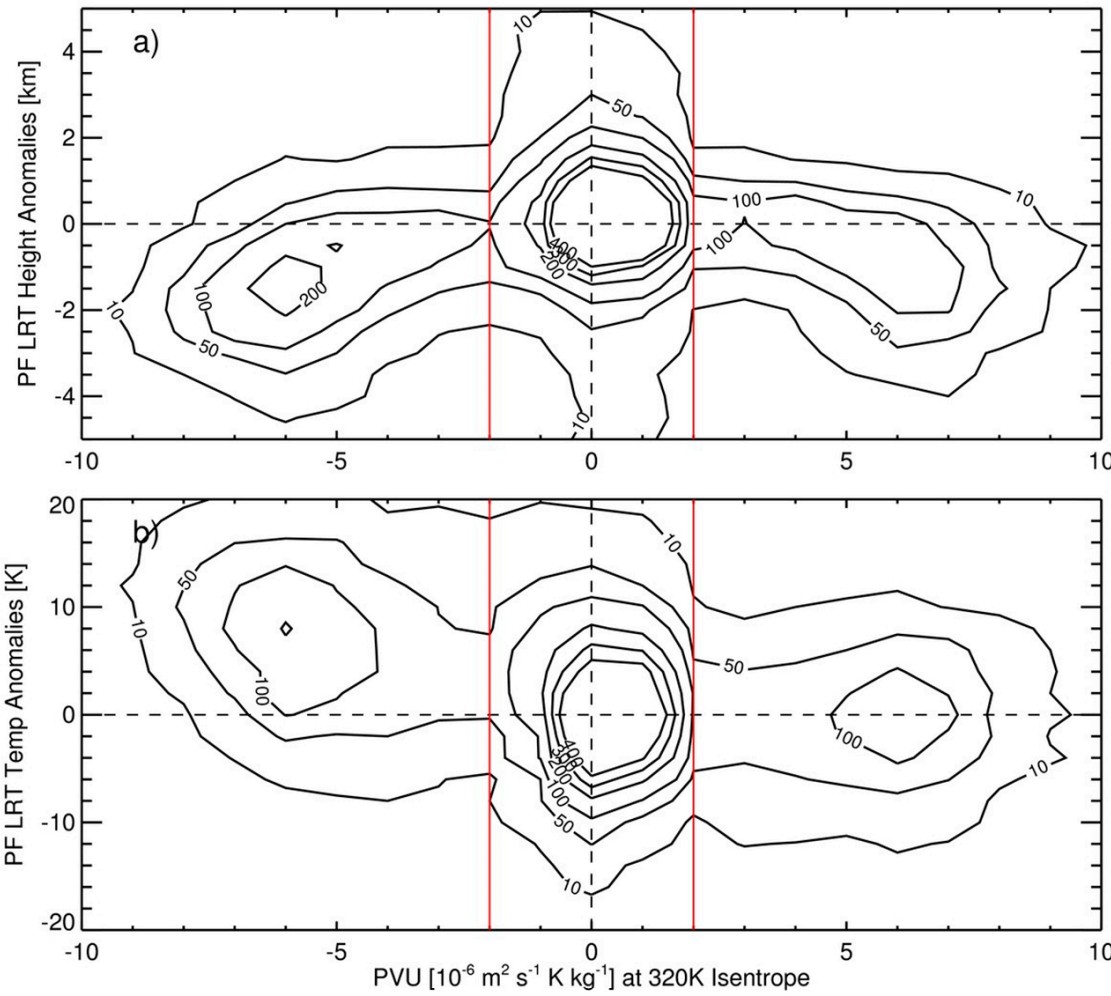

**Figure 4.** Number of PF-collocated GPS-RO temperature profiles comparing the potential vorticity values (PVU) at the 320 K isentrope with the lapse rate tropopause height (**a**, km) and temperature (**b**, K) anomalies that occurred near the PF. Solid red lines (±2-PVU) separate non-DSI and DSI PFs.

## 4. Sampling and PF Characteristics

The extratropics in both the NH and SH (20–65°) are chosen for analysis in this paper. There are 832,858 PFs observed by GPM throughout the extratropics with 10,736 non-DSI PFs and 11,239 DSI PFs with an RD > 0.5 collocated with a GPS-RO temperature profile. Figure 5 displays the distribution of GPS-collocated PFs with a RD > 0.5 for both non-DSI PFs (a,b) and DSI PFs (c,d) categorized by size and RD. Larger-size non-DSI PFs (over 50,000 km$^2$) are most frequently observed over the oceanic regions (Figure 5a). These larger systems are a combination of mesoscale convective complexes, extratropical cyclones that have progressed from mature and into the occlusion stage of development, and tropical cyclones. In contrast, the deepest (overshooting, RD > 1.0) PFs are mainly found over land, such as North America, Europe, and Argentina (Figure 5b). These deepest PFs are primarily summertime features such as squall lines and supercells fueled by warm land surface temperatures and large convective instability. However, some deep PFs also form over warm ocean currents such as the Gulf Stream, some of which are likely tropical cyclones. There are also many shallower non-DSI PFs over both land and ocean, which are mostly comprised of single-cell thunderstorms and warm-frontal precipitation. Also, note that there are few non-DSI PFs in the subtropics off the continental west coasts due to the general lack of precipitation in these regions. On the other hand, DSI PFs (Figure 5c,d) are most often observed in the mid-and-high latitudes, with few occurrences in the subtropics. Most DSI PFs occur over the oceans due to contrasting air masses (baroclinic instability), with the cold and dry continental air masses interacting with the warmer and wetter oceanic air masses. The majority of these PFs include precipitation associated with extratropical cyclones, along frontal boundaries, or within cutoff lows. Several hotspots stand out, such as the wintertime storm tracks over the Southern Ocean, the Icelandic Low, and the Aleutian Low. No consistent pattern is shown for different PF size or RD categories. However, as expected, many more DSI PFs have large sizes, but are rather shallow since extratropical cyclones typically have much weaker uplift than summertime convection.

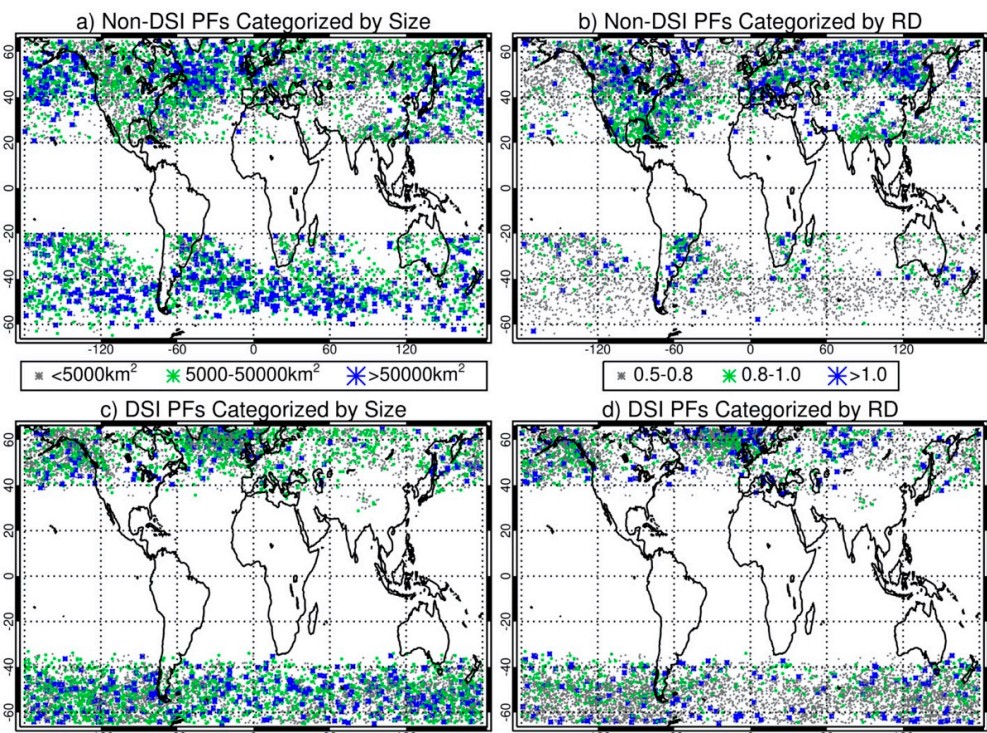

**Figure 5.** Distribution of extratropical PFs (RD > 0.5) observed by GPM from 2014 to 2017 for (**a**) non-DSI PFs categorized by size, (**b**) non-DSI PFs categorized by RD, (**c**) DSI PFs categorized by size, and (**d**) DSI PFs categorized by RD.

Additionally, Figure 6 shows an example of an overshooting convective system (RD = 1.13) observed by GPM in the early morning of 28 July 2015 over South Dakota in the United States. This system is a strong bow-echo, with near-surface reflectivity reaching up to 60 dBZ (Figure 6a) and maximum echo-top heights near 17 km (Figure 6b). A cross-section of the PF is also displayed, with very low brightness temperatures and high rain rates (50 mm h$^{-1}$) occurring near the core of the PF (Figure 6e). The collocated ERA-I PV is near or slightly above 1-PVU, indicating a non-DSI PF (Figure 6c). Finally, the GPS temperature profile near the maximum echo-top height (44.2° N, 101° W) is shown along with its corresponding background profile, and the LRT height and maximum echo-top height are also displayed (Figure 6d). Strong upper tropospheric warming along with cooling around the tropopause are observed for this overshooting PF.

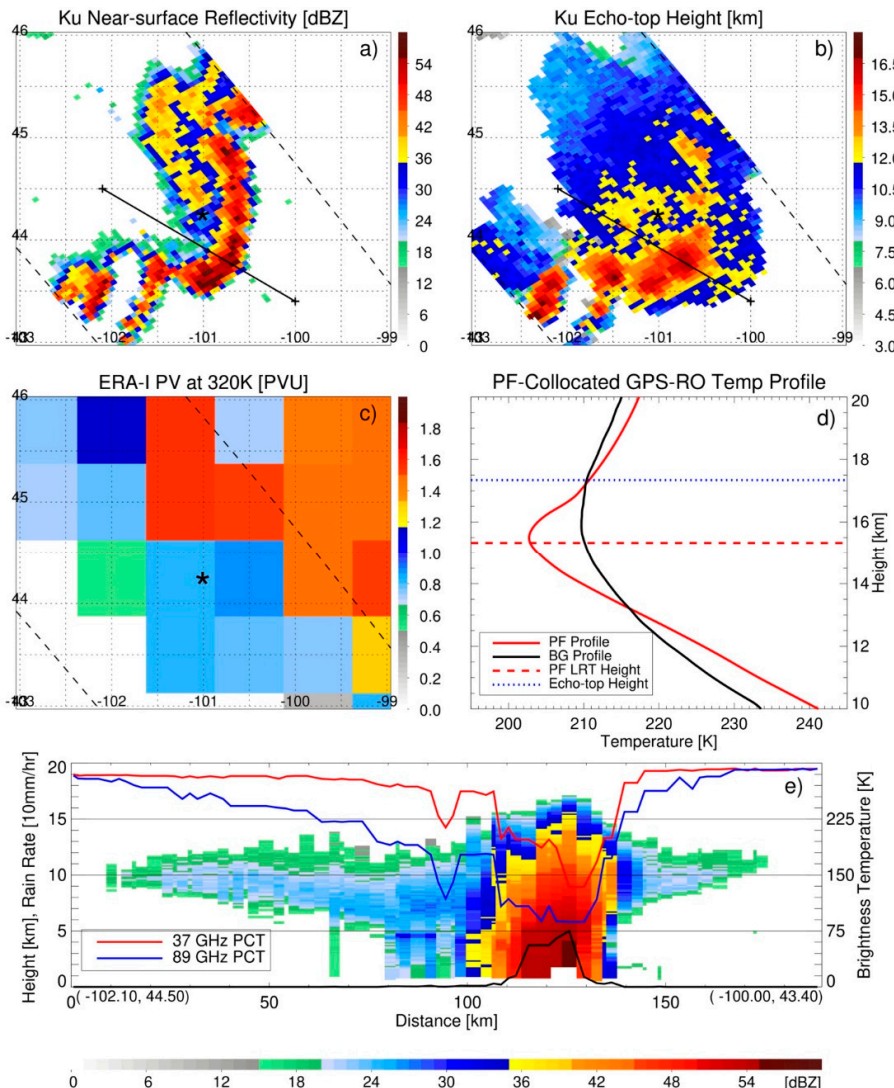

**Figure 6.** Overshooting convection observed by GPM over South Dakota on the morning of 28 July 2015. (**a**) Precipitation radar (PR) Ku-band near-surface reflectivity (dBZ). (**b**) PR Ku-band maximum echo-top heights (km). (**c**) ERA-Interim PV at 320 K potential temperature over the GPM PR swath. (**d**) PF-collocated GPS-RO temperature profile (red) located at 44.2° N, 101° W (denoted by the asterisk in (**a–c**)) and the corresponding background profile (black), along with the PF lapse rate tropopause height (red) and maximum echo-top height (blue). (**e**) Cross-section of PR Ku-band reflectivity (dBZ) through the center of the PF (indicated by the solid black line in (**a,b**)). Red line is the GMI 37 GHz polarization-corrected temperature (PCT) (K); blue line is the GMI 89 GHz PCT (K); and black line is the rain rate (10 mm h$^{-1}$).

## 5. Results

### *5.1. Relationship of PF Characteristics to UTLS Temperature Anomalies*

This section focuses on extratropical PF characteristics and UTLS temperature anomalies within $\pm 5$ km of the PF LRT height. GPS-RO temperature anomalies near both non-DSI and DSI PFs are separated by their surface properties (land or ocean) and further divided into subgroups based on PF size (area) and depth (RD). For example, PFs reaching close to the LRT are grouped together within the "0.8–1.0" RD classification, while PFs reaching above the LRT (overshooting) are grouped within the ">1.0" classification. Finally, PFs are grouped by their month of occurrence to quantify any seasonal differences in anomalies.

#### 5.1.1. Non-DSI PFs

Figure 7 displays the mean GPS-RO temperature anomalies near non-DSI PFs over the extratropics (20–65°). For all PFs classified by various RDs (Figure 7a), warm anomalies are observed for the shallower PFs within roughly 1.5 to 5 km below the LRT, ranging from 0.1 to 0.8 K. Interestingly, the strongest warming (~0.8 K) occurs for the shallower PFs (RD < 0.6), whereas either slight cooling or no anomalies are observed for deeper PFs reaching near or above the tropopause (RD > 1). We speculate that this is because the more intense deep convection typically occurs in the earlier part of a PF's life cycle [65]. As a result, the cumulative influence of latent heat release that occurs around the convective core may not be sufficient to affect the surrounding environment yet [66,67]. Above, the upper tropospheric warming, a strong layer of cool anomalies centered on the PF LRT is observed, ranging from roughly −2.5 to −3 K. Both the shallowest and deepest PFs show comparable magnitudes of strong cooling. Above the cooling layer, warming of 0.5 to 1 K is generally observed with the largest magnitudes for the deepest overshooting PFs. There are major differences in anomaly magnitude between land and oceanic PFs (Figure 7b,c), with much more variation evident for oceanic PFs. For example, oceanic PFs display a wide range of anomalies below the tropopause layer, with moderate warming for the shallower RDs and strong cooling for the deeper RDs. However, warming is observed for all land PF RD categories. Additionally, the largest anomaly magnitudes occur near oceanic PFs, as the strongest tropopause-level cooling, as well as warming below and above the tropopause, is observed in these groups.

Similar temperature anomaly patterns are seen for all PFs with different sizes (Figure 7d). An increase in anomaly magnitude from smaller to larger PF size is observed throughout the vertical extent of the profiles. A warming layer is displayed up to 1.5 km below the LRT for all subgroups, with the largest PFs exhibiting the strongest warming (up to 0.8 K). The strongest cooling at the LRT also occurs for the largest PFs (roughly −3 K). Above the cooling layer, a much smaller range of anomalies is displayed, with minor warming (~0.3 K) observed for all PFs. Differences between land and ocean PF size subgroups are relatively minor (Figure 7e,f), highlighted by slightly stronger warming in both the upper troposphere and lower stratosphere for oceanic PFs. Larger anomaly variation is again observed among different size oceanic PFs, but with much smaller variation compared to the RD dependency.

The UTLS temperature anomalies near extratropical non-DSI PFs show similar patterns to those near tropical deep convection [24], albeit with slightly larger magnitudes. The warm anomalies observed throughout the mid-to-upper troposphere result from latent heat release in-and-near the convective clouds [1,18]. Additionally, some shallower non-DSI PFs are associated with warm frontal precipitation in the extratropics, and latent heat also is released during the formation of different hydrometeor species induced by warm conveyor belts [68,69]. The cool anomalies that occur near the convective tropopause have been characterized in a few different ways. Holloway and Neelin [19] showed that as gravity waves spread convective warming through the free troposphere, hydrostatic pressure gradients will extend above the heating, causing divergence, ascent, and adiabatic cooling aloft. The convective "cold-top" should be thought of as an inherent part of quasi-equilibrium temperature adjustment. Khaykin et al. [21] also discussed two cooling

mechanisms: the first being nonmigrating tides generated by convective diabatic heating; and the second being the systematic injection and turbulent mixing of adiabatically cooled air by cross-tropopause updrafts. Some previous studies have shown that warming occurs in the lower stratosphere, which is also observed in this study. For example, Chae et al. [70] observed warm anomalies at these altitudes and downward motions above the convection. They suggested that strongly divergent flow and turbulent mixing near the cloud tops would mechanically drag the air just above the cloud outward, which would pull comparatively warmer lower stratospheric air down from above the clouds. It has also been shown that fine-scale features in the lower stratospheric static stability field are reminiscent of the planetary wave response to the lower stratospheric diabatic heating associated with convection [71].

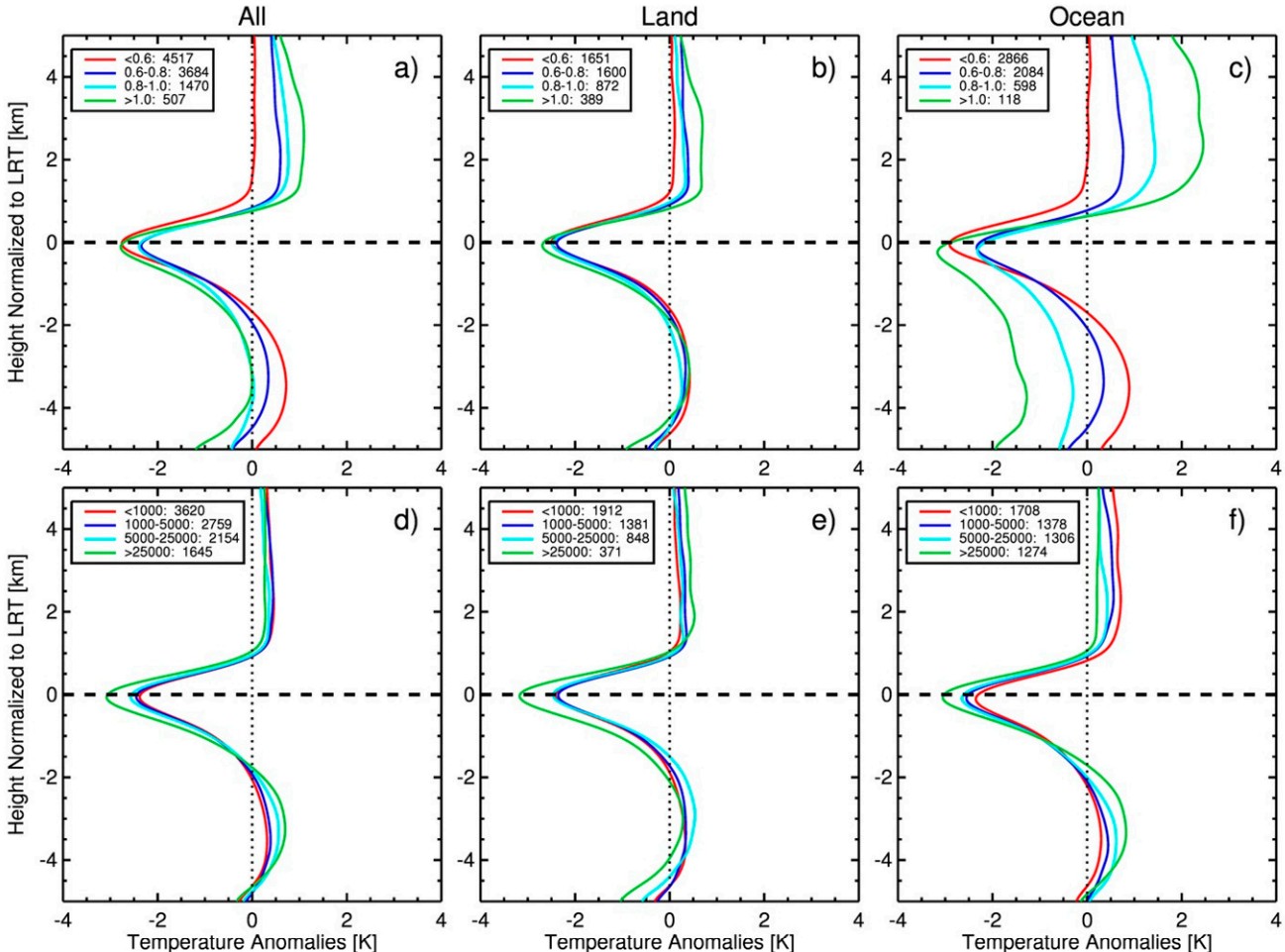

**Figure 7.** GPS-RO temperature anomalies (K) near non-DSI PFs classified by relative depth (**top**) and area (**bottom**, in km²) for (**a,d**) all PFs, (**b,e**) land PFs, and (**c,f**) oceanic PFs. The number of PFs within each category is also displayed after each label.

Figure 8 displays the monthly mean temperature anomalies for non-DSI PFs within the extratropics in both hemispheres. The extratropics are separated into two bands to determine any latitudinal differences (20–40° and 40–65°). Throughout each latitude band, large differences in the pattern and magnitude of anomalies are observed as the seasons change. Within 20–40° N (Figure 8a), the magnitude of upper tropospheric warming remains similar in each month, although the sharpness of the transition from warm to cool anomalies differs. However, tropopause-level cooling is weakest from late spring through early fall and strongest during the wintertime. The magnitude of summertime tropopause-level cooling (between 1.5–2 K) is similar to results shown in previous research

throughout the tropics [24]. Considerable seasonal variation is displayed for the lower stratospheric warming. Similar results are also observed within 20–40° S (Figure 8c). For example, the weakest tropopause-level cooling is again observed during the summer, whereas the strongest occurs during the winter. In contrast, seasonal magnitude changes generally show the opposite patterns in the high latitudes for both hemispheres. Within 40–65° (Figure 8b,d), the strongest tropopause-level cooling is seen during the summer whereas the weakest occurs during the winter. Upper tropospheric warming is slightly stronger in the high latitudes, with the strongest warming occurring during austral summer and boreal winter. We attribute the seasonal magnitude changes in the two latitude bands to seasonal variations in PF characteristics (not shown). Within 20–40° N, the wintertime tends to have more frequent shallow/larger-size PFs relative to the summertime, when more frequent deep convective cores with smaller sizes are observed. However, within 40–65° N, the shallower convection that occurs during the summer has larger areas than the wintertime PFs of similar depth. The SH PFs also display similar characteristics albeit to a smaller degree as the majority of PFs occur over the ocean. This confirms the results from Figure 7, where PFs with larger sizes produced the strongest tropopause-level cooling.

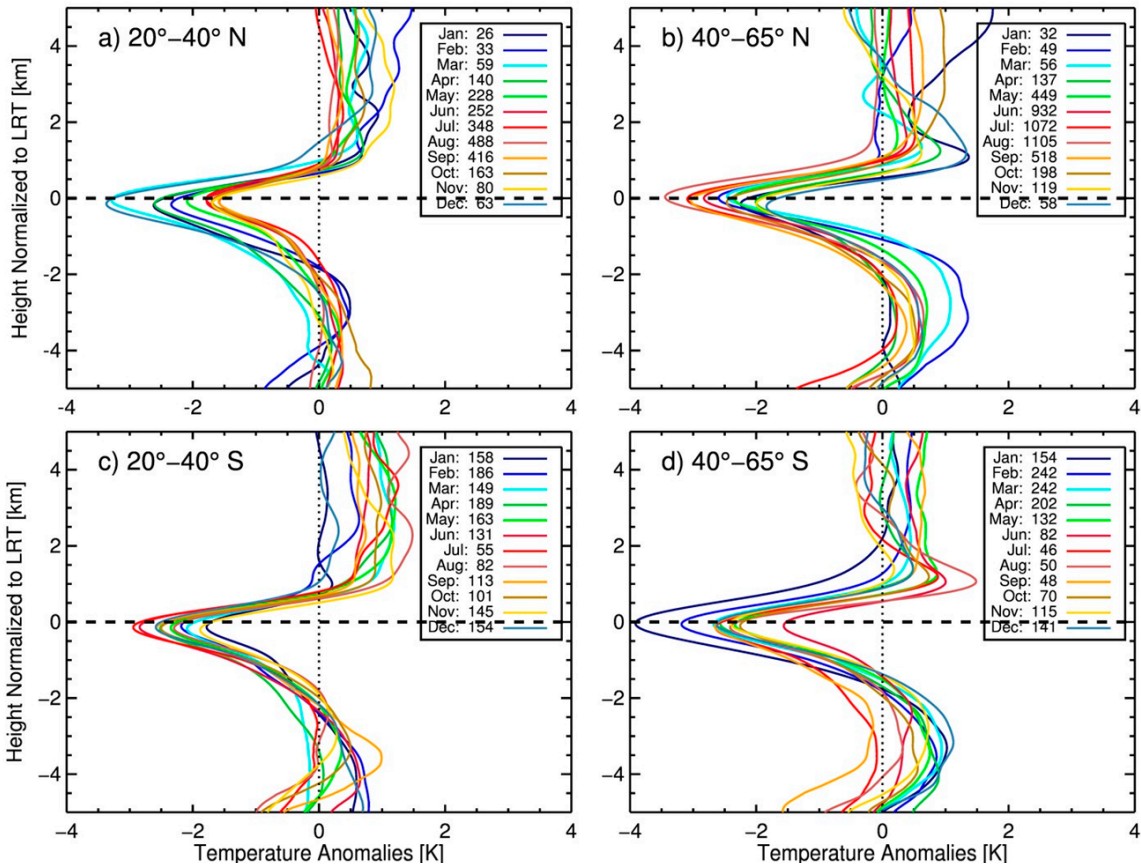

**Figure 8.** Monthly mean GPS-RO temperature anomaly profiles (K) near non-DSI PFs over (**a**) 20–40° N, (**b**) 40–65° N, (**c**) 20–40° S, and (**d**) 40–65° S. The number of PFs within each category is also displayed after each label.

5.1.2. DSI PFs

Figure 9 displays the mean GPS-RO temperature anomalies near DSI PFs within the extratropics (20–65°), with major differences in the anomaly vertical structure observed compared to non-DSI PFs. Near PFs classified by various RDs (Figure 9a), cooling is observed from 5 km below the LRT to roughly 0.5–1 km above the LRT. The cooling noticed throughout the upper troposphere cannot be interpreted by latent heat release from precipitation systems, as it is likely dominantly driven by larger scale dynamic mixing

of baroclinic air masses. Above the LRT, temperature anomalies rapidly transition to strong warming. Maximum warming increases with larger RD (from 2 to 4 K) and occurs at progressively higher altitudes. This warming is largely contributed to by descent in the stratosphere due to large scale mixing associated with extratropical cyclones. Major temperature anomaly differences are evident between land and ocean PFs (Figure 9b,c). Throughout the extratropics, DSI events occur more frequently over the oceans due to the increase in baroclinicity over the warm ocean currents. This stronger baroclinicity is reflected in anomaly magnitudes that are much larger near the oceanic DSI events. On the other hand, the weaker baroclinicity for land DSI events means smaller temperature anomalies are observed. Differences in anomaly structure between land and oceanic PFs with different sizes (Figure 9e,f) are less pronounced, with oceanic PFs again displaying larger anomaly variation between different sizes.

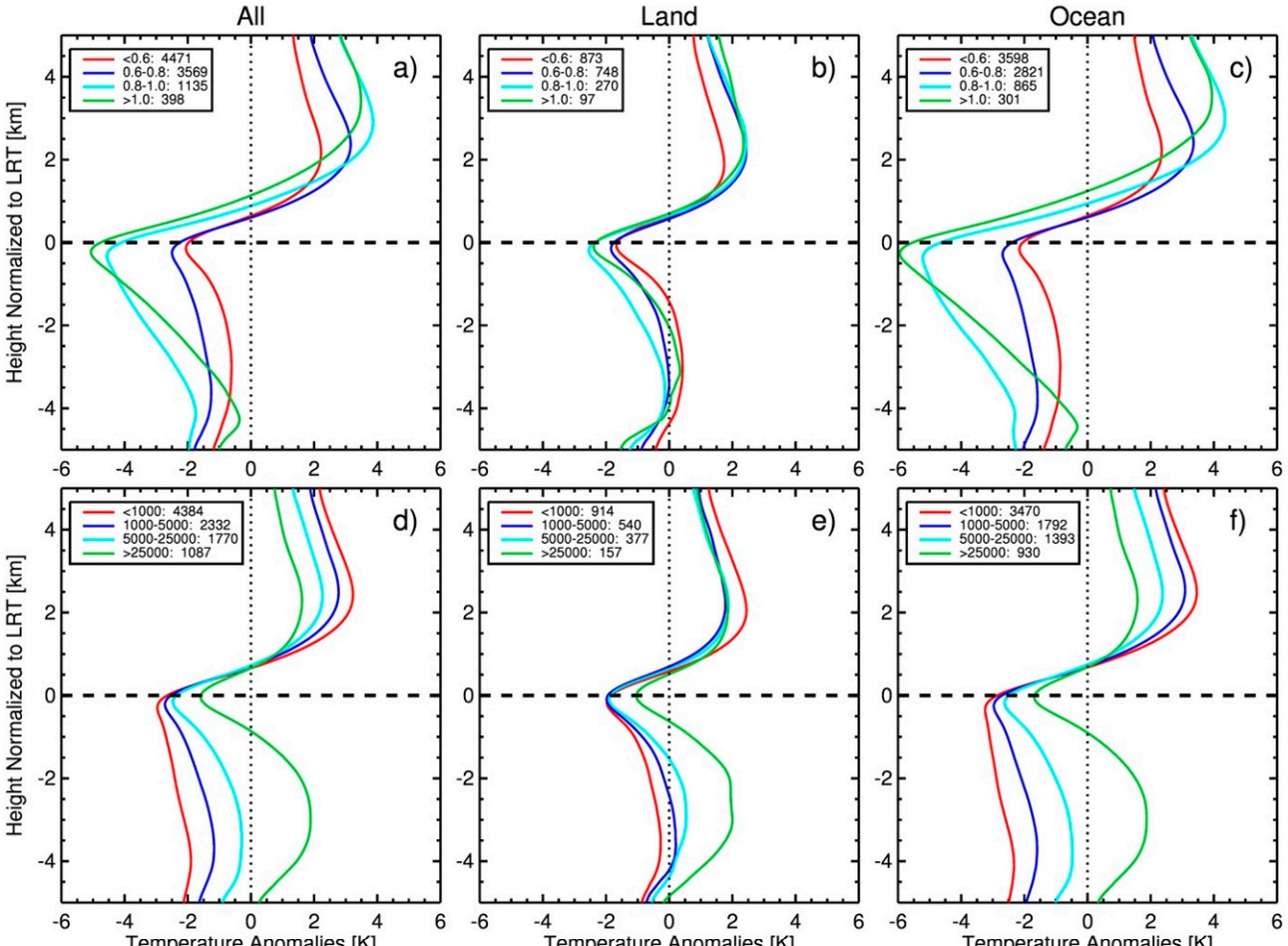

**Figure 9.** GPS-RO temperature anomalies (K) near DSI PFs classified by relative depth (**top**) and area (**bottom**, in km$^2$) for (**a,d**) all PFs, (**b,e**) land PFs, and (**c,f**) oceanic PFs. The number of PFs within each category is also displayed after each label.

It is worth noting the UTLS temperature anomalies (and tropopause anomalies shown later) observed near DSI PFs are strongly influenced by extratropical dynamics rather than just the PFs themselves (unlike non-DSI PFs). Extratropical cyclones obtain their energy from sharp horizontal temperature and moisture differences between tropical and polar air (e.g., highly baroclinic environments). We speculate that the differences in the magnitude of temperature anomaly variation as DSI PF area and RD increase are related to the maturing and dissipation process in these extratropical cyclones. During the early stages of extratropical cyclone development, surface frontogenesis is accompanied by strong differential temperature advection of cold and warm air. The PFs are often driven by

large thermodynamic instability, with a stronger intensity, reaching higher altitudes, and having relatively smaller sizes. Thus, UTLS temperature anomalies are largest at this stage due to the stronger baroclinicity. As the cyclone matures, the system becomes occluded, thereby growing in size, but weakening in strength. The continued dynamic mixing among the two contrasting air masses would then be displayed as weaker cooling in the upper troposphere (Figure 9a,d). After the cyclone has mixed away the temperature contrasts across the front, there is no longer any potential energy available to keep the cyclone going so it dissipates. On the other hand, the larger, more mature systems have ample time to spread latent heat release to the surrounding upper tropospheric environment (evidenced in the upper tropospheric warming displayed only near the largest PFs). However, these speculations would need further validation through additional observations and model simulations in the future. Thermal advection also plays a role in the observed temperature anomalies for DSI events, but quantifying this role would be far from straightforward. For example, strong frontal boundaries in highly baroclinic environments have a mixture of warm and cold air advection. These boundaries would need to be analyzed separately, their direction of motion verified, and the location of the GPS-RO profile determined relative to this motion (as the collocation can be within a 300 km radius of the PF centroid) to better appreciate how thermal advection impacts the results.

DSI cases are associated with tropopause folding, where deep frontogenetic forcing can occur commonly near and upstream of precipitation events in baroclinic environments [72]. A tropopause fold is an extrusion of stratospheric air within an upper-tropospheric baroclinic zone that slopes downward from the normal tropopause level to the middle or lower troposphere [73]. The fold is associated with a substantial lowering of tropopause heights and has been identified to be a contributor to upper-level frontogenesis and the rapid development of surface storms [72,74]. In summary, the DSI PFs themselves are not the main cause of the UTLS temperature anomalies observed and are more directly related to the nearby dynamical processes.

Figure 10 displays the monthly mean temperature anomalies for DSI PFs within 40–65° in both hemispheres. For DSI events, even larger seasonal differences are observed in the pattern and magnitude of anomalies as the seasons change. In the NH, wintertime events display cool anomalies throughout the entirety of the upper troposphere, with maximum cooling of −4 K occurring near the tropopause. Additionally, lower stratospheric warming is also at a maximum in the winter (~3 K). In contrast, summertime events display weak warming in the upper troposphere (~0.5 K) along with relatively weaker cooling (less than −2 K) near the tropopause and warming (~1 K) in the lower stratosphere. Anomaly magnitudes during the transition seasons generally remain in between summer and winter magnitudes. Similar results are also shown for the SH. Cool anomalies are observed throughout the entirety of the upper troposphere in all seasons, but with stronger summertime cooling near the tropopause (−2 to −3 K). The larger-amplitude wintertime anomalies are likely due to stronger temperature contrasts fueling the development of intense midlatitude cyclones, which cause deeper tropopause folding and significant stratosphere-troposphere exchange. These contrasts weaken significantly in the summer, leading to reduced dynamic UTLS mixing, which results in anomaly profiles that look more similar to non-DSI PFs. Additionally, the minor differences in anomaly structure displayed between the NH and SH are in contrast to previous results from Biondi et al. [75], which observed considerable structural differences in anomalies between NH and SH extratropical cyclones.

### 5.2. Relationship of PFs to Tropopause Height/Temperature Anomalies

In this section, we focus on the changes that occur specifically to the thermal tropopause near PFs. Two-dimensional histograms are constructed relating the RD and size of the PF to the magnitude of tropopause height and temperature anomalies observed. The PFs are binned using intervals of 0.1 for RD and bin sizes increase logarithmically from 100 km$^2$

to ~120,000 km² for area. Contours of the number of PFs within each bin are also plotted (excluding the bins with fewer than ten profiles).

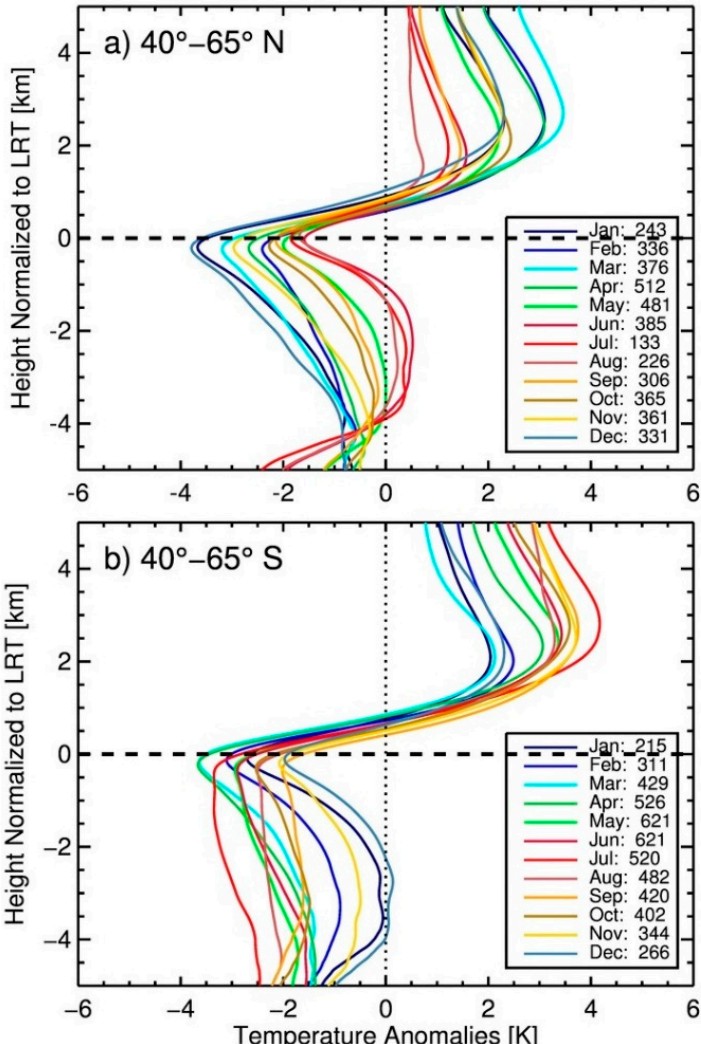

**Figure 10.** Monthly mean GPS-RO temperature anomaly profiles (K) near DSI PFs over (**a**) 40−65° N and (**b**) 40−65° S. The number of PFs within each category is also displayed after each label.

Figure 11 displays the LRT height and temperature anomalies for non-DSI and DSI PFs within the extratropics (20−65°). For non-DSI PFs, moderate LRT height decreases are observed (0.1 to 0.4 km), on average, for PFs with a high RD (Figure 11a). In contrast, considerable LRT height increases (0.2 to 0.6 km) are observed for PFs with a low RD, and magnitudes increase as PF size increases. The corresponding temperature anomalies (Figure 11c) show a similar pattern, but with the opposite sign compared to the height anomalies. On the other hand, DSI events are consistently associated with negative LRT height anomalies (Figure 11b) along with corresponding positive LRT temperature anomalies (Figure 11d). Much larger anomaly magnitudes are observed relative to non-DSI PFs. Anomaly magnitude is larger near PFs with a stronger RD, which likely occurs in the earlier stages of cyclone development. The largest LRT height anomalies (−1.5 to −2.0 km) and temperature anomalies (6 to 8 K) are observed near the highest RDs, whereas the smallest LRT height anomalies (−0.25 to −0.5 km) and temperature anomalies (0.1 to 2 K) are observed near the largest PFs with a shallow RD, when cyclones are likely in the occlusion stage.

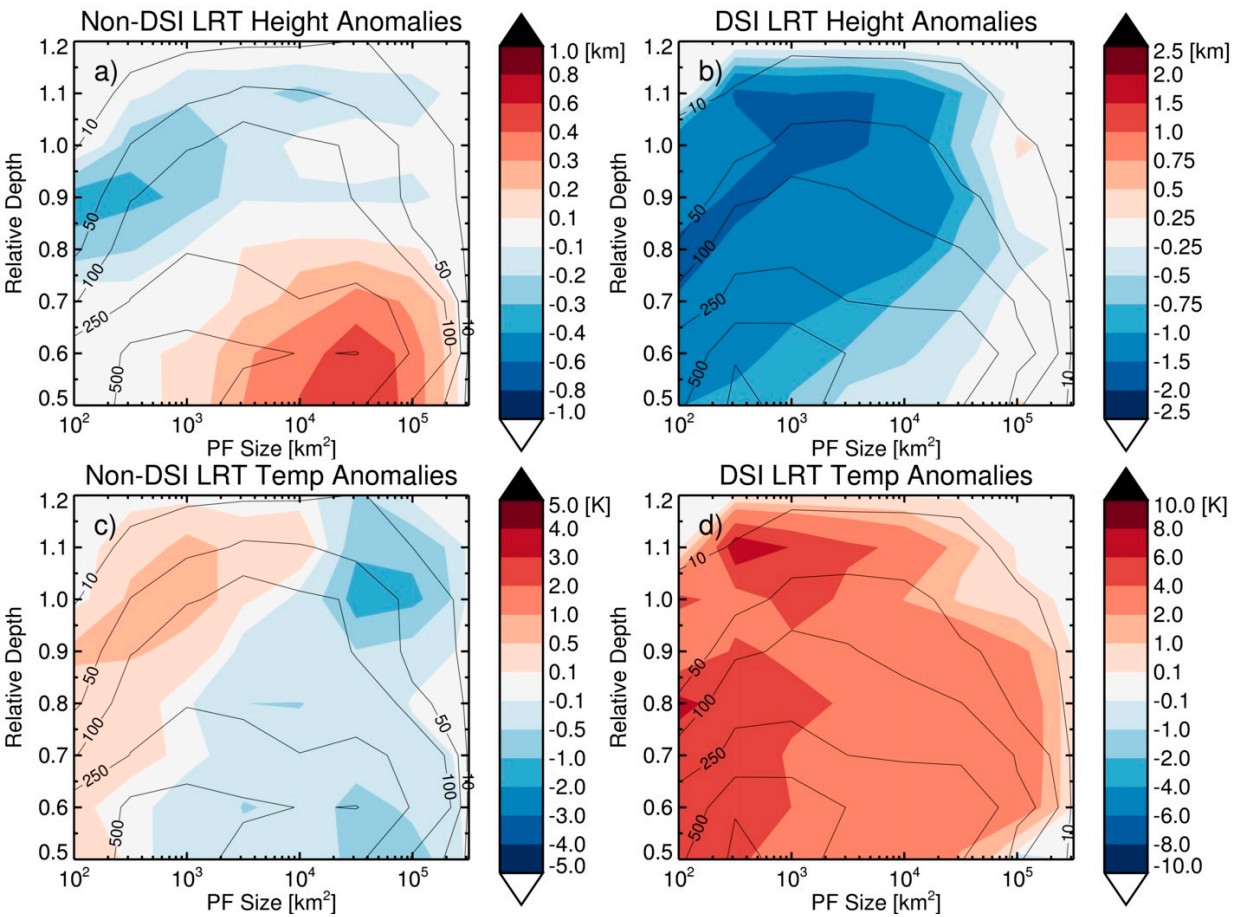

**Figure 11.** Two-dimensional histogram (solid contours) and joint distribution of lapse rate tropopause height and temperature anomalies (shaded contours) with corresponding PF size and relative depth for non-DSI (**a**,**c**) and DSI (**b**,**d**) PFs. Note the color bar scale differences in each panel.

Figure 12 displays the observed LRT temperature/height anomalies (solid contours) along with the temperature anomalies at the PF LRT relative to the background profile at the same altitude (the anomalies at the "zero line" from Figures 7 and 9; shaded contours), as well as individual cases representing typical RO profiles near each type of PF. Non-DSI PFs throughout the extratropics (Figure 12a) display a wide range of tropopause anomalies, with LRT temperature anomalies up to ±12 K and height anomalies up to ±2.5 km. However, most of these samples are clustered near the middle with only a slight trend toward the lower-right quadrant. Most PFs are typically found in the upper-left and lower-right quadrants, which indicates that the tropopause is most commonly raised/cooled or lowered/warmed. Even though the PF tropopause displays considerable variation relative to the background tropopause, strong cooling (shaded contours) is almost always observed for the PF LRT *relative to the background temperature at the same altitude* (displayed in Figure 7). The strongest cooling (between −8 to −10 K) occurs when the PF LRT is pushed much higher (e.g., 2 km), but notable cooling is also observed (between −2 to −6 K) even when the PF LRT is at a lower altitude than normal. This is explained by a commonly-observed type of profile seen in the lower-right quadrant and is shown in Figure 12c, where a PF-collocated GPS profile (red) is compared to its background profile (black). In this instance, the PF LRT is almost 1 km lower and 3 K warmer than the background LRT. However, note that the PF LRT is roughly 2 K colder than the background temperature at this altitude. Near the top of the PF (blue dotted line), the lapse rate increases compared to the background and a relative minimum temperature is observed above the convective cloud. Above this relative minimum, an inversion commonly occurs, and then

the temperature progresses towards another relative minimum at a slightly higher altitude. This often results in a double tropopause structure, as a new, lower LRT forms near the cloud top and the second (higher) LRT remains either near or a few kilometers above the climatological LRT, similar to previous results shown by Biondi et al. [76]. While this type of profile was observed throughout the entire extratropics, it was most commonly seen in the subtropics. Additional research into the formation of these "lower" tropopauses near the convective top is recommended, especially for determining how long this type of UTLS environment persists after PF occurrence.

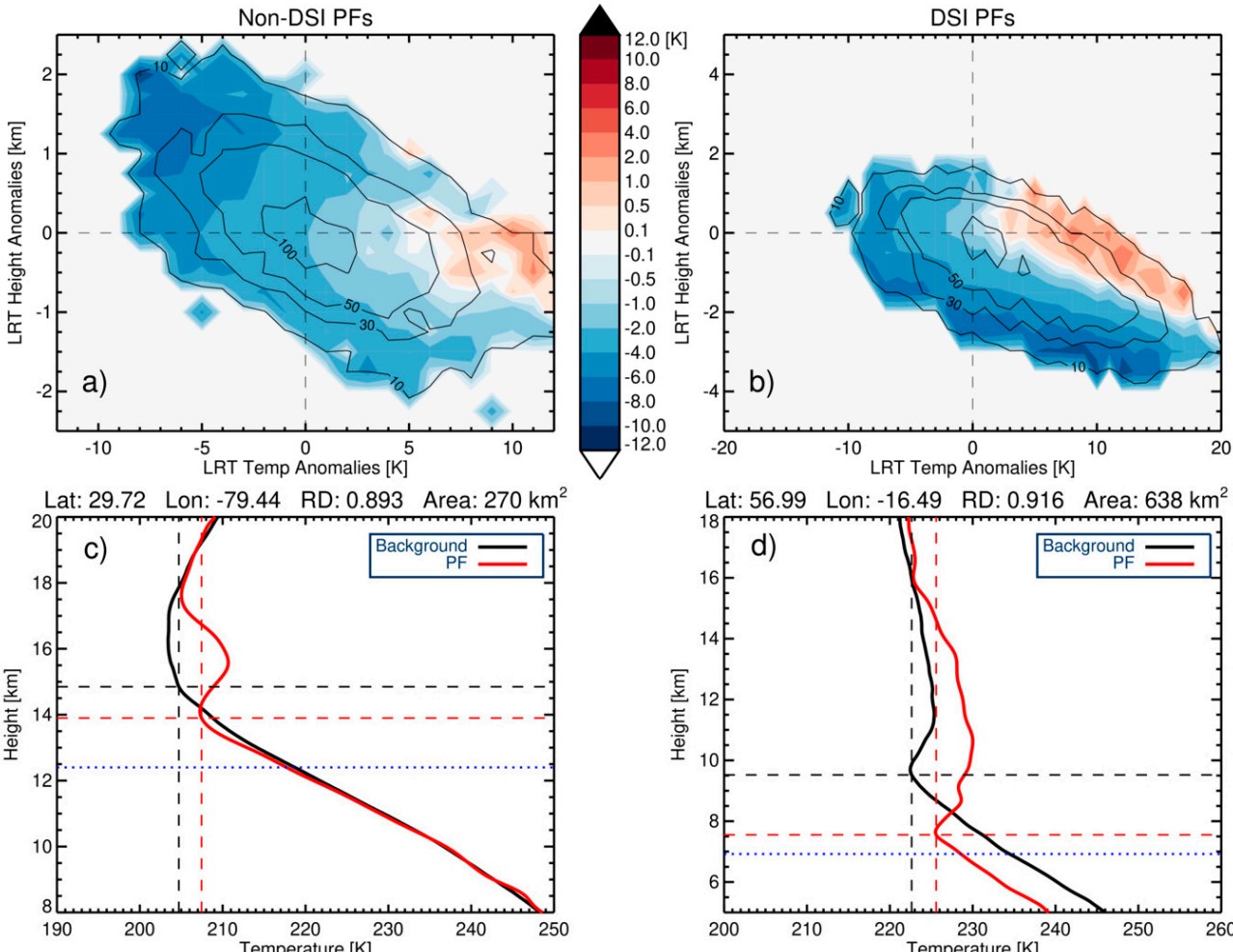

**Figure 12.** Two-dimensional histogram of the PF LRT temperature (K) and height (km) anomalies (solid contours) and the joint distribution of the temperature anomalies (K) at the PF LRT from Figures 7 and 9 (shaded contours) for (**a**) non-DSI and (**b**) DSI PFs. Note the different axis ranges in each panel. Also shown are examples of common PF-collocated GPS temperature profiles (red) for (**c**) non-DSI (5 July 2015) and (**d**) DSI (2 May 2016) PFs along with their associated background profiles (black). Dashed lines indicate the tropopause height and temperature for both profiles and dotted lines indicates maximum echo-top height.

Similarly, DSI PFs throughout the extratropics (Figure 12b) display a wide range of tropopause anomalies. The amplitude of maximum anomalies nearly doubles for DSI events, with LRT temperature anomalies up to 20 K and LRT height anomalies near −4 km. Most samples are within the lower-right quadrant, indicating the PF LRT typically becomes much lower and warmer. Nevertheless, similar to non-DSI PFs, strong cooling is again observed at the PF LRT relative to the background temperature at the same altitude. However, the strongest cooling now occurs when the PF LRT height decreases the

most. A typical GPS temperature profile for a high-RD DSI PF is displayed in Figure 12d. The PF temperatures are much colder than the background environment throughout the troposphere. The cold UTLS air descends deeply into the troposphere along a tropopause fold, which is associated with typical baroclinic features such as strong cold fronts and extratropical cyclones [31]. The PF LRT shows a much sharper temperature transition, with a height about 2 km lower (at ~7.5 km) and temperature ~3 K warmer than normal. The tropopause inversion layer, which is a region of enhanced static stability above the extratropical tropopause associated with a narrow-scale temperature inversion [77], occurs between ~7.5 to 10.5 km and becomes somewhat stronger. Finally, temperatures remain above normal throughout the lower stratosphere near the PF. This type of profile is the most common profile observed near DSI PFs, although there are some instances when the PF LRT can be pushed higher and become colder.

## 6. Discussion and Conclusions

In this paper, the relationship between extratropical precipitation and the thermodynamic structure of the UTLS in both hemispheres (20–65°) was studied by collocating precipitation features observed by the GPM precipitation radar with high vertical resolution GPS-RO temperature soundings from 2014 to 2017. PFs were classified into two different categories (non-DSI and DSI) using PV. UTLS temperature and tropopause height/temperature anomalies were calculated and evaluated according to PF characteristics (e.g., PF sizes, depths, surface types, and seasonality).

For non-DSI PFs, warm anomalies ranging from 0.1 to 1 K were observed from roughly 5 to 1.5 km below the LRT due to latent heat release in-and-near the convective clouds. This transitioned to a layer of strong cooling ranging from −2 to −3 K centered around the PF LRT height, which occurs due to the injection of adiabatically cooled air near the tops of the PFs. Beginning about 1 km above the PF LRT, minor-to-moderate warming ranging from 0.2 to 1 K was observed. Furthermore, a consistent progression in anomaly magnitude was observed for upper tropospheric warming as well as cooling around the PF LRT as PF size increased. In contrast, the RD displayed mixed results on anomaly magnitude, as the strongest warming in the upper troposphere and cooling near the PF LRT generally occurred for PFs with the shallowest RD. We speculate this may be related to the life cycle of a PF, as intense overshooting convection typically occurs in the earlier stages of convective development and enough time may not have passed to spread the warming and cooling to the environment surrounding the PF.

On the other hand, DSI events displayed distinctly different patterns as the UTLS temperature response was dominated by large-scale dynamics, with strong cooling ranging from −2 to −5 K (except near large-size PFs) observed throughout the mid/upper troposphere and maximum values occurring slightly below the PF LRT. The anomalies transitioned to moderate/strong warming above the LRT, ranging from 2 to 4 K. The strongest upper tropospheric cooling and lower stratospheric warming occurred near the deepest PFs, whereas the weakest magnitudes were observed near the largest size PFs, which we attributed to properties of PFs under the different stages of extratropical cyclone development. The temperature anomaly pattern observed during DSI events likely occurred due to strong horizontal dynamic mixing among contrasting tropical and polar air masses and contributes to the large temperature standard deviation observed throughout the upper troposphere within the mid/high latitudes. Additionally, larger magnitudes and more variation in the temperature anomalies was evident for oceanic DSI events, which was expected as baroclinicity is typically strongest near the coasts.

In addition, seasonal differences in anomalies for both PF categories were explored. For non-DSI PFs between 20–40° in both hemispheres, tropopause-level cooling was weakest from late spring through early fall and strongest during the wintertime. However, within the higher latitudes (40–65°), seasonal magnitude changes showed the opposite patterns, as the strongest tropopause-level cooling was seen during the summer, whereas the weakest occurred during the winter. This was attributed to latitudinal seasonal variations of PF

characteristics. For DSI events within 40–65° in both hemispheres, wintertime PFs displayed cool anomalies throughout the entirety of the upper troposphere and the strongest near-tropopause cooling of any season. In contrast, minor warming was observed in the upper troposphere along with weaker near-tropopause cooling for summertime events. The larger-amplitude wintertime anomalies likely occurred due to stronger temperature contrasts fueling the development of intense midlatitude cyclones.

Finally, the relationship between extratropical PFs and LRT height/temperature was quantified relative to PF area and RD. For non-DSI PFs, minor decreases in LRT height (−0.1 to −0.4 km) and increases in LRT temperature (0.1 to 1.0 K) were generally observed for PFs with a deeper RD, whereas PFs with a shallower RD and large area displayed large LRT height increases (0.2 to 1 km) and temperature decreases (−0.1 to −2 K). Overall, non-DSI PFs throughout the extratropics displayed a wide range of tropopause anomalies, with LRT temperature anomalies up to ±12 K and height anomalies up to ±2.5 km. Two types of UTLS temperature profiles were commonly observed for non-DSI PFs: (1) the PFs either push the tropopause higher and it becomes colder; or (2) a double tropopause structure develops, with a new, lower tropopause forming near the top of the convective cloud beneath the higher climatological LRT. DSI events were associated with much larger tropopause anomalies. On average, LRT heights were 0.25 to 2 km lower and LRT temperatures were 2 to 8 K warmer than normal during significant deep stratospheric intrusion events, indicative of subsidence and tropopause folding. Additionally, LRT height decreases of almost −4 km and temperature increases near 20 K were observed. There was one common profile type observed near DSI PFs, as the LRT displayed a much stronger than normal temperature transition and a robust tropopause inversion layer was seen.

It is worth noting that there are some limitations in this study. First, GPM only provides a snapshot of the PF at the time of observation so it cannot capture the stage of PF development (although it can be inferred to some extent). This additional information would be useful for providing a more in-depth explanation of the differences in the temperature anomaly magnitudes under different PF sizes and depths. Second, UTLS temperature and tropopause anomalies would ideally be determined relative to the larger neighboring environment of each PF. Unfortunately, this is not possible with GPS-RO sounding density, so a composite study is unable to describe the full influence of convection on its environment. Third, the number of GPS-RO soundings available in the extratropics has steadily declined in recent years, as satellite missions such as COSMIC have reached the end of their life cycle. The recently launched COSMIC-2 mission only provides soundings within the tropics and subtropics, so additional GPS-RO satellites are needed to provide necessary coverage throughout the extratropics. However, despite these limitations, these results suggest intricate relationships between different types of precipitation systems and the UTLS temperature structure and tropopause throughout the extratropics. Importantly, non-DSI PF temperature anomaly patterns and magnitude agreed well with previous research on tropical convection, which we believe provides a "unification" of tropical and extratropical deep convection impacts to UTLS temperatures.

This study enhances understanding of extratropical precipitation and UTLS temperature responses, and it is hopeful that this will lead to the continued improvement of precipitation system representation in weather models. Additional studies on the role that PFs play in moisture transport in the extratropical UTLS would also be beneficial in understanding extratropical STE and will likely be a topic of future research.

**Author Contributions:** Conceptualization, B.R.J., F.X. and C.L.; Methodology, B.R.J., F.X. and C.L.; Software, B.R.J.; Validation, B.R.J.; Formal analysis, B.R.J., F.X. and C.L.; Investigation, B.R.J.; Resources, F.X.; Data curation, B.R.J.; Writing—original draft preparation, B.R.J.; Writing—review and editing, B.R.J., F.X. and C.L.; Visualization, B.R.J.; Supervision, F.X.; Funding acquisition, F.X. All authors have read and agreed to the published version of the manuscript.

**Funding:** This research was funded by the National Science Foundation, grant number 2054356; and by the National Aeronautics and Space Administration, grant numbers NNX14AK17G and NNX15AQ17G.

**Institutional Review Board Statement:** Not applicable.

**Informed Consent Statement:** Not applicable.

**Data Availability Statement:** All datasets used in this study are freely and openly available. Thanks to the Precipitation Processing System (PPS) team at NASA Goddard Space Flight Center, Greenbelt, MD, for GPM data processing assistance. GPM precipitation feature database is available at http://atmos.tamucc.edu/trmm/ (accessed on 10 January 2019). GPS-RO data was obtained from CDAAC at UCAR: https://data.cosmic.ucar.edu/gnss-ro/ (accessed on 10 January 2019). ERA-Interim data was obtained from ECMWF: https://apps.ecmwf.int/datasets/data/interim-full-daily/levtype=pl/ (accessed on 10 January 2019).

**Acknowledgments:** The authors would like to thank Bill Randel, Thomas Winning, and Kevin Nelson for helpful discussions on the project.

**Conflicts of Interest:** The authors declare no conflict of interest and the funders had no role in the design of the study; in the collection, analyses, or interpretation of data; in the writing of the manuscript, or in the decision to publish the results.

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
