# Peer review of "Relationships between Extratropical Precipitation Systems and UTLS Temperatures and Tropopause Height from GPM and GPS-RO"

_atmosphere, doi:10.3390/atmos13020196_

Round 1
Reviewer 1 Report
The manuscript presents relationships between extratropical precipitation systems and UTLS temperature as well as tropopause temperature and height for two types of precipitation systems: deep stratospheric intrusion (DSI) and non-deep stratospheric intrusion (non-DSI) events. While data from the GPM satellite are used to detect precipitation features (PF), ERA-Interim data are used for their classification. GPS RO temperature profiles are finally used to investigate the thermal structure of the atmosphere close to precipitation events.
The authors use 4 years of data (2014 to 2017) and analyze the thermal atmosphere structure of DSI and non-DSI PFs. Temperature anomalies are investigated over land and over the ocean as well as for different seasons. The influence of relative depth and PF size is also analyzed.
In general, the paper is well and clearly written but the methodology is not clear to me. Most figures are appropriate and descriptive.
Major comment:
Methodology: Section 3 is worse organized and methodology not clear.
- I strongly recommend reorganizing this section and adding a flowchart, which summarizes main steps.
- Remove Section 3.1 (and Fig. 1), which is rather a motivation for the study than part of the methodology.
- It is not clear to me if “background” profiles are obtained only from GPS profiles collocated to PF, or if “all” GPS profiles enter this statistics.
- Figure 2 shows GPS RO sampling data from 2006 to 2017 but PFs are only observed from 2014 to 2017. Does this make sense?
- Quality control (i.e., profiles, which are excluded from the study) should be described early in this section.
- The influence of subtropical double tropopauses needs to be addressed. These double tropopauses occur throughout the year and might have a strong impact on the results. Fig. 12c, e.g., shows a typical example of a subtropical temperature profile. I am not convinced that the vertical structure of this temperature profile is only caused by the PF.
General comments:
Acronyms: Introduce all acronyms the first time and stick to using the acronym afterwards (mainly revise the introduction).
Data sets: Please add some more information about the data sets. What is the horizontal resolution of the data sets? What are their vertical resolutions? Give some information about the GPS RO tangent point trajectory. What are the data set’s error characteristics?
ERA-Interim: The ERA-Interim data set is outdated. Why did the authors not use the more recent ECMWF reanalysis ERA5?
DSI PF events: There are only very few DSI PFs in the subtropics. Actually, there are hardly any between 20°S/N and 40°S/N (Figure 5c and d). Why? Please add an explanation and consider discussing statistics only between 40°S/N and 65°S/N (e.g., in Fig. 12).
RD classification is inconsistent. While Fig. 5 shows RD classes 0.5-0.74, 0.75-0.99, and >=1, Figs. 7 and 9 show RD classes <0.6, 0.6-0.8, 0.8-1.0, and >1. Please use a consistent classification throughout the manuscript and revise the plots accordingly.
Discussion of results, lines 500 to 513: This discussion is confusing and needs to be rewritten. At the end, the authors write “Previous studies have shown both warming and cooling in the layer above the tropopause, which is also observed in this study” (lines 508 to 509). What do the authors refer to? I always see a distinct cooling at tropopause level and a warming approximately 1 km above.
Specific comments:
Line 12: What does “anomalies” refer to? Please clarify and add this information in the manuscript text.
Line 13: “Two background environments are established”: The authors probably refer to DSI and non-DSI events. However, this is not entirely clear because they are introduced much later in the abstract. Please clarify.
Line 64: Please add some references
Lines 109 to 113: “These two environments are separated because PF development is similar in nature within each environment, as non-Deep Stratospheric Intrusion precipitation features form mainly due to convective instability whereas precipitation near Deep Stratospheric Intrusion events typically forms from baroclinic instability.” I do not understand why you separate the environments, if PF development is similar within each environment. Please clarify.
Lines 174/175: The additional use of Metop data would have increased sampling density of GPS RO measurements because Metop provides significantly more measurements than TerraSAR-X and GRACE-B. Is there any reason why the authors decided not to use this data set?
Line 186: The correct units of k1 and k2 are K/hPa and K2/hPa, respectively.
Line 189: dry pressure from radio occultation measurements is not defined as pressure without water vapor. It rather is the pressure, which is derived from refractivitiy but neglecting the second term of the Smith-Weintraub equation (i.e., shifting the contributions of humidity into temperature and pressure). Please clarify and rewrite this statement.
Lines 201/202: ECMWF stands for “European Centre for Medium-Range Weather Forecasts”
Line 217: “upper bound on” --> “upper bound of”
Line 240: introduce “NH” after “Northern Hemisphere”
Line 243: “during the winter” --> “during the NH winter”
Line 245: “during the winter” --> “during the NH winter”
Line 246: “in the spring” --> in the Southern Hemisphere (SH) spring”
Line 274; “late fall through early spring”: Is this true for NH and SH spring? Please clarify.
Line 286: “most frequently in fall, winter, and spring, depending on latitude”: Please give more information about where and when
Figure 2: GPS RO sampling might refer to the number of GPS RO profiles going into the statistics. The number of GPS RO profiles decreases with height. At which level did you extract these numbers?
Lines 335 to 343: In my opinion, this information would better fit into Section 2.1.
Lines 353 to 358: Do not repeat the figure caption in the manuscript text.
Line 402 & 403: “a RD” --> “an RD”
Figure 5: Between 20°S/N and 30°S/N, there is clear gap of non-DSI events close to the continental west coasts. Please add an explanation in the manuscript text.
Figure 6a, b, and c: Move the x- and y-axis labels out of the figure for better readability
Figure 6d and line 435: What is the mean tangent point of the collocated GPS RO temperature profile?
Figure 6: Add a note that Figure 6e shows the cross section indicated in Figure 6a and b (black solid line)
Lines 460 to 461: “For all PFs classified by various RDs (Figure 7a), warm anomalies are observed for PFs within roughly 1.5 to 5 km below the LRT, ranging from 0.1 K to 0.8 K.” This statement is not correct. Temperature anomalies are predominantly negative for RD 0.8-1.0 and >1.0.
Lines 462 to 464: “Interestingly, the strongest warming (~0.8 K) occurs for the shallower PFs (RD<0.6) whereas weak warming occurs for deeper PFs reaching near or above the tropopause (RD>1).” I do not see any warming for deeper PFs (RD>0.8) below ~0.75 km. Please clarify.
Lines 474/475: “moderate warming for the shallower RDs and strong cooling for the deeper RDs.”: Do you have any explanation for this feature?
Line 487: “cooling at the LRT for oceanic PFs”: do you mean “less cooling at the LRT for oceanic PFs”?
Line 535: “the shallower convection….have” --> “The shallower convection….has”
Lines 560 to 562: “In general, oceanic PFs still display larger anomaly variation between different sizes, but only on the order of 1-2 K compared to 2-4 K for different RDs.”: This statement is not true for the upper troposphere.
Figure 11: Add a note that different colorbars are used in each panel.
Figure 12: Add a note that different x- and y-axes are used in each panel.
Figure 12a and 12b: There are hardly any DSI events between 20°S/N and 40°S/N (see comment above). How does this statistics look like for 40°S/N to 65°S/N?
Author Response
Please see the attached Word document for detailed responses to each comment.

Reviewer 2 Report
Please see the attached file.

Author Response

(The authors gave the same response as above.)

Reviewer 3 Report
Dear Authors, please find the review comments in the uploaded pdf.

Author Response

(The authors gave the same response as above.)

Reviewer 4 Report
This paper classifies precipitation features as non-deep stratospheric intrusion (non-DSI) or deep stratospheric intrusion (DSI) by potential vorticity and characterizes the relationship between extratropical precipitation systems and upper troposphere and lower stratosphere (UTLS) temperature and tropopause height anomalies within different environments. The results clearly demonstrate different responses of the tropopause under these two classes. The conclusions are meaningful and the manuscript is well written. I suggest acceptance as it is.
Author Response

(The authors gave the same response as above.)

Reviewer 5 Report
An interesting study with an overall clear presentation of the results. I recommend publication after minor revision.
Minor comments:
Title: I think UTLS is common enough of an acronym to use in a title, which makes it easier to read
L93-94: "2014" is not really "now" any more
L93: introduce GPM again
L108: introduce DSI and non-DSI and use acronym consistently from then on
L130: start new line with "Extratropical.." to improve readability
L186: the units of k1 and k2 are "K hPa^-1" and "K^2 hPa^-1", respectively
L200: why not use the newer ERA5 data?
L271: I think a figure showing isentropes and PV for latitude vs altitude would be useful. the authors could e.g. show only 4 panels in Fig 1 (4 individual months, or 4 seasons) and add 4 panels showing isentropes and PV.
L307: the authors should use southward and northward instead of equatorward and poleward, since that is how it shifts in the SH.
L319-326: in my opinion, these sentences ("In addition, ....nearly 60 degree N.") can be removed, and a new paragraph is started with "In order to minimize.."
L347: correct and change sentence to: "..is derived by subtracting the respective DSI or non-DSI gridded median GPS background temperature profile (TBG) from the GPS PF temperature (TPF) profile"
L351: change sentence to: "Figure 3 shows two examples of GPS temperature profiles collocated with PFs (red) and their gridded median non-DSI and DSI background profiles (black)"
L358: change sentence to: "Then, the anomalies are grouped according to the different PF characteristics (PF size, RD values) and .."
L385-390: change to: "Non-DSI PFs have most samples clustered to minor LRT anomalies, although a few LRT height/temperature anomalies can reach ±5 km/±20 K. The majority of DSI events, on the other hand, display negative LRT height anomalies, generally along with a corresponding LRT temperature increase. Anomalies are larger in the SH than in the NH."
Figure 6: move latitude and longitude labels and panel titles outside the figure area so they can be read. Please indicate the location of the RO PF profile in panels a and b. It would be helpful to add the resepective units to the colorbars.
Figure 8: Consider showing seasonal means, or only 1 month per season. 12 lines make it actually quite hard to see the described patterns.
L623: small, medium, and large are not good descriptors of the PF size. You could relate it either to previously used area bins (as in Figure 5), or define it here.
Author Response

(The authors gave the same response as above.)
